

# The impact of small-scale surface representation in WRF on hydrological modeling in a glaciated catchment

Florentin Hofmeister[1,2], Xinyang Fan[3,4], Madlene Pfeiffer[5], Ben Marzeion[5,6], Bettina Schaefli[3], and Gabriele Chiogna[4]

[1]Bavarian Academy of Science and Humanities, Geodesy and Glaciology, Munich, Germany
[2]Chair of Hydrology and River Basin Management, Technical University of Munich, Munich, Germany
[3]Institute of Geography and Oeschger Centre for Climate Change Research, University of Bern, Bern, Switzerland
[4]Department of Geography and Geosciences, GeoZentrum Nordbayern, Friedrich-Alexander-University Erlangen-Nuremberg, Erlangen, Germany
[5]Institute of Geography, University of Bremen, Bremen, Germany
[6]MARUM – Center for Marine Environmental Sciences, University of Bremen, Bremen, Germany

**Correspondence:** Xinyang Fan (xinyang.fan@unibe.ch)

**Abstract.** High-elevation alpine catchments are particularly affected by the global rise in temperature. Understanding the drivers of climate-induced changes in the hydrological response of these catchments in the past is relevant for developing future adaptation strategies for water resources and risk management. However, the study of long-term changes since the last Little Ice Age (around 1850) is strongly limited by the availability of hydrometeorological observation data. Regional climate models (RCMs) can bridge this limitation and provide comprehensive meteorological forcing data for hydrological models (HM). We used the Weather Research & Forecasting Model (WRF) to dynamically downscale a global reanalysis product (20CRv3) to a 2 km x 2 km spatial and 1 h temporal resolution from 1850 to 2015 as forcing for an HM (WaSiM). The main challenge is transferring the forcing data to the much finer grid resolution (i.e., 25 m) of the HM, considering the complex topography and plausible sub-daily precipitation and temperature lapse rates (TLRs). Thus, we developed a workflow for extracting and transferring hourly TLRs from the WRF atmosphere to the small-scale topography of the HM domain. In addition, we corrected WRF precipitation frequencies with observation data and re-distributed the precipitation according to the small-scale topography. Our study demonstrates the impact of TLRs computed from different WRF layers (i.e., 2 m and free atmosphere) on the HM results of a highly glaciated Alpine catchment in the European Alps. In a multi-data evaluation procedure, we found that the TLRs and the HM results are significantly dependent on the coarse surface properties of WRF. Temperature-sensitive processes such as snow and glacier evolution, as well as the streamflow response, are more realistically simulated when the HM is forced by TLRs originating from the WRF free atmosphere rather than with simulated near-surface temperature. The HM results are also consistent with observation data over a simulation period beginning in 1969, suggesting the corrected WRF temperature can reliably reproduce the non-stationarity in local temperature observations. Our study addresses several aspects, limitations, and potential solutions in applying a standard modeling chain of an RCM and a physics-based HM for climate sensitivity studies in high-elevation alpine regions.





# 1 Introduction

The European Alps are particularly affected by climate change, as the observed temperature increase is approximately twice the global average, meaning an average warming of 1.8 °C since 1880 (Kotlarski et al., 2023). However, this trend has not been linear since pre-industrial times, as warming has significantly accelerated in the last 30 years (Nigrelli and Chiarle, 2023),
with considerable consequences for the entire Alpine geosystem and its inhabitants. Consequently, geomorphological activity increases (Altmann et al., 2020, 2024) due to dramatic glacier retreat (Beniston et al., 2018; Sommer et al., 2020) as more and more loose sediments are available for fluvial transportation in proglacial areas (Piermattei et al., 2023). Moreover, the accelerated warming rates of the European Alps are probably leading to heavier rainfall intensities and consequently to greater and sudden floods (Wilhelm et al., 2022).

Seasonal snow evolution is also highly affected by the increase in temperature in the European Alps. Snow cover duration (Rumpf et al., 2022; Notarnicola, 2022) and snow depth (Matiu et al., 2021; Marcolini et al., 2017, 2019) have declined in the last 30 years, which has considerable economic implications for ski tourism (Morin et al., 2021) and hydropower production (Magnusson et al., 2020). Investigating snow droughts in the Alpine region, i.e., the deficit of snow water equivalent (SWE), and its consequences on the entire hydrological system are increasingly becoming the focus of the scientific community (Chiogna
et al., 2018; Basilio Hazas et al., 2022; Brunner et al., 2023). The risk of particularly severe socio-hydrologic droughts can increase significantly due to the co-occurrence of snow and summer drought, such as the one that occurred in northern Italy in 2022 (Avanzi et al., 2024). Although rich in fresh water, the European Alps could lose their status as water towers in already dry regions due to the decline in snow and glacier meltwater and the simultaneous increase in summer evapotranspiration (Mastrotheodoros et al., 2020). This directly affects the Alpine population and the people living along the major Alpine rivers
(Immerzeel et al., 2020).

The study of climate-induced long-term changes mainly depends on station observations. However, meteorological observations are limited in spatiotemporal resolution, especially since the last Little Ice Age (LIA) around 1850. Homogenized temperature and precipitation records are available before 1850 (Auer et al., 2005, 2007), but they are not equally distributed and mainly represent lower-laying areas of the Alpine region (Tiel et al., 2022). The linear extrapolation of these tempera-
ture records to higher elevations is problematic because several studies have found different trends in the elevation-dependent warming (e.g., Rangwala and Miller, 2012; Kuhn and Olefs, 2020). High-resolution Regional Climate Models (RCMs) bridge this spatiotemporal gap by dynamically downscaling global reanalysis products to a much finer grid resolution of a few kilometers. To account for the complex orography of the Alps, multiple studies have proven the benefit of high-resolution RCM for regional climate change studies in the European Alps (Gómez-Navarro et al., 2015; Warscher et al., 2019). A frequently used
model for long-term atmospheric simulations is the Weather Research and Forecasting (WRF) model (Powers et al., 2017). The main advantage of dynamical downscaling with atmospheric models is the physical consistency, which means that all fields are downscaled simultaneously, so the interdependencies between downscaled variables are resolved (Berg et al., 2024). The major disadvantage is the high computational effort to run the WRF model: Even with high-performance computing, centennial



simulations for an area of approximately 24,416 km$^2$ with 2 km spatial and 1 hour temporal resolution can take up to several
months.

Another recent approach for dynamical downscaling of meteorological variables is made with the High-resolution Inter-
mediate Complexity Atmospheric Research (HICAR) model (Reynolds et al., 2023), which is a variant of the existing ICAR
model, developed specifically for simulations down to the hectometer scale while maintaining relatively low computational
costs (Berg et al., 2024). Especially the lower computational time compared to WRF makes the HICAR model a more efficient
solution to problems with multiple model runs at high resolutions and large spatial extents (Berg et al., 2024). Berg et al.
(2024) were able to demonstrate for the first time the effects of applying dynamical downscaling schemes to physics-based
snowpack simulations, including wind- and gravitational-induced snow transport, at the seasonal and catchment scale. Their
results indicate the importance of accurate meteorological forcing data and snow redistribution schemes to match observed
spatial snow accumulation patterns and snow height distribution.

Although many high-resolution WRF climate simulations have a spatial resolution of a few kilometers (Gómez-Navarro
et al., 2015; Warscher et al., 2019; Velasquez et al., 2020), this is still far too coarse for resolving localized precipitation pat-
terns or small-scale processes such as snow redistribution in high-elevation areas (Berg et al., 2024). Moreover, the simulated
near-surface heat fluxes depend on the spatial resolution of the WRF elevation model and the resolution and accuracy of land
cover classes (i.e., land-atmosphere interactions). In addition, elevation-dependent processes, such as temperature and precip-
itation lapse rates, are not sufficiently resolved by the RCMs on the sub-kilometer scale due to the coarser topography, which
often under-represents terrain features. Comprehensive correction procedures are therefore necessary to rescale and correct the
coarse-scale RCM data to a much finer spatial resolution in mountainous regions, as usually required by distributed hydro-
logical models (HM). Velasquez et al. (2020) developed a three-step procedure for bias-correcting WRF precipitation over a
complex terrain in the Swiss Alps. This approach explicitly considers orographic characteristics. The first step is the separation
of precipitation concerning different orographic characteristics, the second is the adjustment of daily precipitation with very
low intensity, and the third is the application of empirical quantile mapping (EQM). EQM consists of adjusting the quantile
values from a simulation to those from the observations through a transfer function (Velasquez et al., 2020). Therefore, the
EQM method is based on the availability of meteorological time series with sufficient temporal coverage and quality. However,
meteorological observations are particularly limited in mountainous areas, especially in relation to reliable precipitation records
from high altitudes (> 2000 m a.s.l.) (Foehn et al., 2018). In combination with the aspect of non-stationarity in meteorological
observations (Teutschbein and Seibert, 2012), the evaluation of bias correction methods in data-scarce mountainous regions is
particularly challenging.

HMs can serve as an additional validation method for the reliability of bias corrections, as adjusted forcing data will affect
the quality of HM results (Teutschbein and Seibert, 2012; Shrestha et al., 2017; Hakala et al., 2018; Hofmeister et al., 2023).
It can be assumed that a bias correction must also lead to more realistic and plausible HM results compared to a model forcing
based on the uncorrected RCM (Teutschbein and Seibert, 2012). However, the focus should not only be on good and plausible
streamflow results for alpine catchments, but also on the reliability of the other hydrological processes (Hofmeister et al.,
2023). Mainly, elevation-dependent processes such as snowfall and melt generation depend on a reliable representation of



temperature and precipitation lapse rates in mountainous regions (Hakala et al., 2018; Hofmeister et al., 2023; Berg et al.,
2024). Although most of the previous evaluations of bias correction methods with HMs focused on monthly or daily model
results (Teutschbein and Seibert, 2012; Shrestha et al., 2017; Hakala et al., 2018), Faghih et al. (2022) found that even the diel
cycle of variables simulated by RCMs is also biased, which raises issues about the necessity of correcting such biases prior
to streamflow simulations at the subdaily timescale. They recommend correcting the diel cycle for hydrological simulations to
accurately represent the diel cycle of summer streamflow in small catchments (i.e., < 500 km$^2$) (Faghih et al., 2022).

From our perspective, three primary research questions arise that address general and highly relevant aspects of alpine
hydrology. i) What impact do small-scale surface processes of RCM have on distributed hydrological models? ii) How to
transfer coarse-resolved RCM data to higher spatial resolutions required by distributed hydrological models, considering the
complex topography of high mountain areas? and iii) How consistent are the bias-corrected RCM data over a longer period (>
30 years) with respect to non-stationary conditions (i.e., temporal homogeneity)?

In this study, we demonstrate the variability and efficient correction of WRF near-surface temperatures concerning eleva-
tion and land cover classes in a highly glaciated catchment of the eastern European Alps. We evaluate the spatiotemporal
variability of the uncorrected and corrected WRF temperature with a multi-data approach that includes a fully distributed and
physics-based HM (WaSiM). Our study provides comprehensive and valuable insights into the complex process interdepen-
dency of the cryo- and hydrosphere of highly-glaciated catchments and the need for an accurate spatiotemporal temperature
and precipitation representation in such a rugged topography for climate sensitivity studies.

## 2   Material and methods

### 2.1   Study site

The research site is located in the upper Kauner Valley in the Tyrolean Alps (Austria), has an altitudinal range from 1810
to 3535 m a.s.l. (Altmann et al., 2020) and is mainly drained by the Fagge River, a tributary of the upper Inn River system.
The drainage area of the Fagge at the stream gauge Gepatschalm (GA), south of the Gepatsch reservoir (see Fig. 1), covers
about 54 km$^2$. The actual drainage in this study is a bit larger (62 km$^2$), as it extends beyond the stream gauge GA and
the catchment is closed at the inflow to the Gepatsch reservoir. The streamflow regime of the Fagge is highly impacted by
snow and glacier melt, which results in a distinct seasonality with low flows in winter and high flows in summer. The glacier
coverage is relatively high with a 30 % fractional cover in 2015 (Buckel and Otto, 2018), but has declined from 39 % in 1969
according to the first Tyrolean Glacier Inventory (Patzelt, 2013). Accordingly, the entire geosystem of the upper Kauner Valley
has undergone profound changes in the respective period, which were investigated in multiple studies (e.g., Altmann et al.,
2020, 2024; Piermattei et al., 2023). With several existing long-term hydrometeorological observation stations (see Fig. 1),
the upper Kauner Valley is an ideal study site for HM applications (e.g. Förster et al., 2016; Rogger et al., 2017; Pesci et al.,
2023). In 2020, scree (vegetation cover < 20 %) was the dominant land cover at 32 %, closely followed by glacier cover at 27
%. Rocks with vegetation cover about 12 % of the upper Kauner Valley. Subalpine coniferous forest only covers 1.3 % of the
research area. The land cover classification was carried out by Ramskogler (2023).





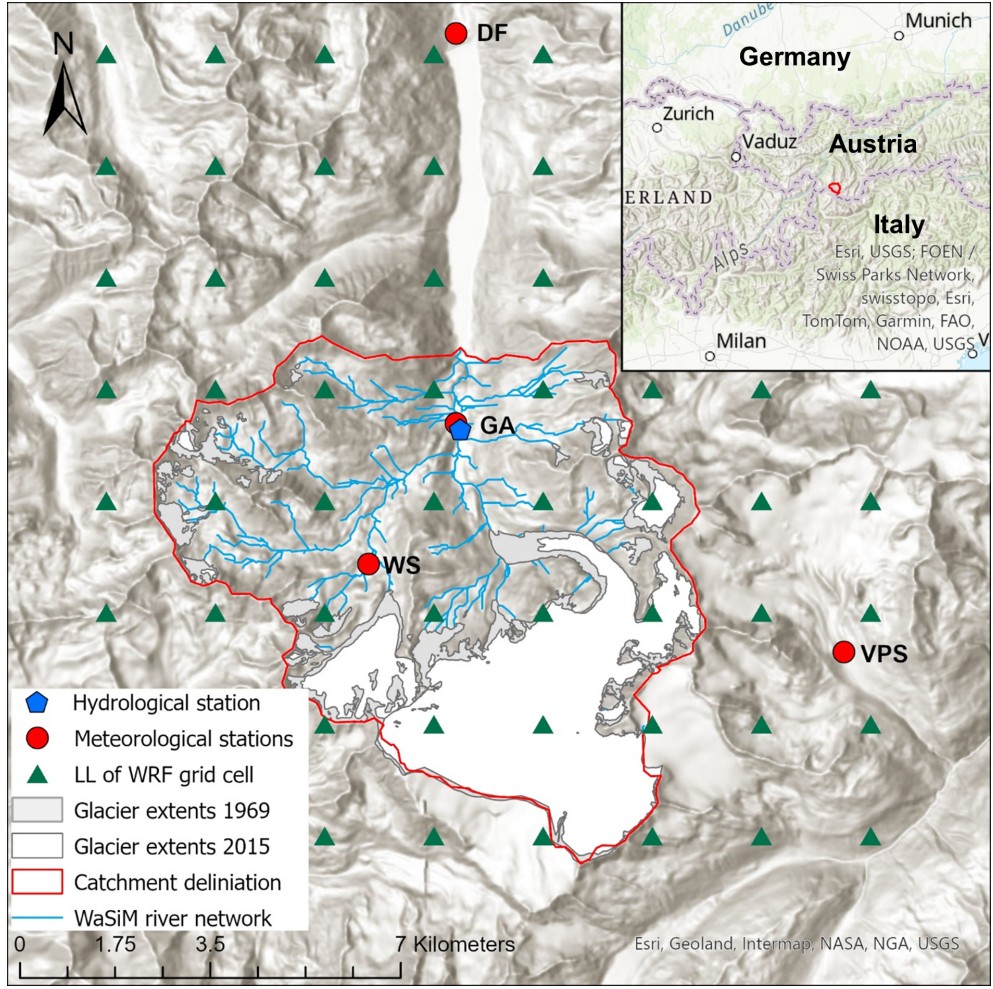

**Figure 1.** Overview map of the research area upper Kauner Valley in Tyrol (Austria). Green triangles illustrate the lower left coordinates of the 2 km WRF grid cells. Meteorological stations are indicated with red circles. Hydrological stations are represented with a blue symbol. Glacier extents are plotted for 1969 from the first Tyrolian Glacier Inventory (GI 1) and 2015 from the fourth inventory (GI 4). Catchment delineation and river network were computed based on the official digital elevation model of Tyrol from 2006, which is available from the Province of Tyrol (URL in the *Code and data availability* section).

## 2.2 Hydrometeorological observation data

High-resolution (i.e., 15 minute intervals) streamflow data at gauge Gepatschalm (GA) were provided by TIWAG-Tiroler Wasserkraft AG, the local hydropower plant operator, for the period 1971 to 2022. Larger data gaps of several months mainly affect the first four years of streamflow observation at GA. Most of the meteorological stations in upper Kauner Valley (Dammfuss, Gepatschalm, Weisssee), listed in Table 1, are also operated by TIWAG. The TIWAG temperature time series includes only a low fraction (i.e., <1 %) of data gaps. The Bavarian Academy of Science operates multiple stations at the nearby Ver-



**Table 1.** Metadata of the meteorological stations and statistics about the data availability of temperature observations. The coordinates are given as UTM coordinates for zone 32N (epsg code: 32632).

| Station | Station code | East | North | Elevation (m a.s.l.) | Start | Amount of NA | Share of NAs (%) |
|---|---|---|---|---|---|---|---|
| Dammfuss | DF | 632260.0676 | 5202225.106 | 1770 | 1988 | 844 | 0.36 |
| Gepatschalm | GA | 632251.1494 | 5195058.382 | 1900 | 2009 | 18 | 0.03 |
| Weisssee | WS | 630648.4005 | 5192468.238 | 2540 | 2006 | 15 | 0.03 |
| Vernagt Pegel | VPS | 639386.0800 | 5190853.371 | 2640 | 1975 | 24550 | 7.00 |

nagtferner. The main station at Vernagt Pegel has recorded hourly meteorological variables since 1975 and is the highest station (2640 m a.s.l.) with the longest temporal coverage in the region. Due to some failures of the station and sensor, the temper-
ature time series has about 7 % of data gaps for the period 1975-2015. Not listed in Table 1 is the meteorological station in Vergoetschen (1280 m a.s.l.) that HD-Tyrol operates in the lower Kauner Valley. This station was chosen to correct the precip-itation frequency of the regional climate model (WRF) due to its high reliability and long temporal coverage (since 1971) of daily precipitation measurements. As all stations are still in operation, only the start year of the observation is given in Table 1.

### 2.3 Snow observation data

The snow observation data are used to evaluate the simulation results of the HM at the point scale (with station-based snow water equivalent (SWE) data) and at catchment scale with binary snow cover data retrieved from an optical remote sensing product (i.e., MODIS).

#### 2.3.1 Calculation of SWE data

Snow depth (SH) measurements within the research site were only available at the TIWAG station Weisssee since 2007. To
transform daily SH to daily SWE, we applied the recently developed deltaSnow model (Winkler et al., 2021). SWE is modeled by a process-based multi-layer approach considering empirical regression models based on the relationship between density and diverse at-site parameters (Winkler et al., 2021). The deltaSnow model (version 1.0.2) is available in the "*nixmass*" R package and takes continuous daily SH records without data gaps as input. We used the same model parameters as published by Winkler et al. (2021).

#### 2.3.2 Snow cover data

Daily and nearly cloud-free MODIS snow cover maps are available for the entire European Alps in 250 m grid resolution since summer 2002 (Matiu et al., 2019a, b). Despite the relatively coarse grid resolution compared to other optical snow products (e.g., Landsat, Sentinel 2), MODIS snow cover maps can provide valuable information about the seasonal snow cover evolution in high temporal resolution (i.e., daily). Due to the high temporal resolution and global coverage, snow cover data
from MODIS has already been used in many studies focusing on hydrological modeling in high-elevation catchments (e.g.,





Duethmann et al., 2014). After resampling and cropping the MODIS snow cover maps to the respective catchment resolution (i.e., 25 m) and domain, we set glaciated raster cells to no data as they are permanently snow-covered and would bias the comparison with the snow cover simulations of the HM. We use glacier extents from the third Tyrolean glacier inventory from 2006 (Abermann et al., 2012). To ensure consistency between observed and simulated snow cover maps, we set the same grid

cells to no data in both products and normalize the fractional snow-covered area (fsca) values from 0 (no snow) to 1 (completely snow-covered).

## 2.4 Dynamical downscaling with WRF

The model forcing of the HM is generated with the Weather Research and Forecasting (WRF) model (version 4.3), which is based on fully compressible and non-hydrostatic equations (Skamarock et al., 2008). The global Twentieth Century Reanalysis

version 3 (20CRv3) data set (Compo et al., 2011; Giese et al., 2016; Slivinski et al., 2019), with a spatial and temporal resolution of 1 ° × 1 ° and 3 h is used as driving data (initial and boundary conditions). The simulation is performed in three nested domains, with a grid spacing of 18 km (domain 1), 6 km (domain 2), and 2 km (domain 3). The simulated data in domain 3 are stored at a temporal interval of 1 h. The vertical discretization is based on 49 virtual atmospheric layers, the so-called eta levels, which allows the WRF model to resolve atmospheric flow near complex terrain more accurately. The Yonsei University

scheme is selected to calculate the planetary boundary layer (Hong et al., 2006). For further information on the WRF model configuration, refer to Altmann et al. (2024). The considered WRF grid cells of this case study cover an area of about 200 km$^2$, as also parts of the neighboring valleys are included (Fig. 1).

We use the Noah land surface model (LSM), which is one of the most widely used land surface components of WRF and plays a crucial role in simulating interactions between the land surface and the atmosphere. The LSM calculates the energy and

mass transfer at the ground interface, providing bottom boundary conditions to the atmospheric model (Tomasi et al., 2017), in this case WRF. The Noah LSM is based on 24 land use categories originating from the United States Geological Survey (USGS) (Anderson et al., 1976). From the 24 land cover categories, only seven are represented in the WRF domain of the upper Kauner Valley (Fig. S1). Namely, *Woodland Mosaic*, *Grassland*, *Shrubland*, *Evergreen Needleleaf*, *Wooden Wetland*, *Wooded Tundra*, and *Snow or Ice*. The most frequent category of the WRF Kauner Valley domain is *Wooded Tundra* (15 grid cells),

followed by *Snow or Ice* (11 grid cells). The latter category means permanent snow- and ice-covered areas where snow and ice cover persist for at least ten months of the year, typically exceeding 60 % of the area. This category signifies a significant presence of snow and ice as a permanent landscape feature (Anderson et al., 1976). The category *Snow or Ice* covers most of the higher-elevation research site (i.e., > 2900 m.a.s.l. in Fig. S1), including the Gepatsch glacier.

Figure 2 shows the relation between grid cell surface elevation (m) and simulated temperatures averaged over a test period

of three months (i.e., from August to November) stratified according to the land cover per grid cell. The temperatures shown correspond to different eta levels, the 2 m above ground temperature (T2) and the surface (or skin) temperature (TSK). Eta level 1 (Fig. 2c) corresponds on average to an altitude of about 17 m above the ground surface over the three-month test period. It is computed by subtracting the mean WRF geopotential height of eta level 1 from the mean WRF surface height (2785 m) of the upper Kauner Valley. Eta level 49 (Fig. 2f) has an average altitude of approximately 30 km above the ground





surface. The TSK and T2 elevation plots (Fig. 2) show much lower temperatures for snow and ice-covered grid cells and a certain scatter, which shows that these simulated temperatures are strongly influenced by surface properties, with a snow-related surface cooling effect present up to T2. The elevation gradients (both for snow and no snow) of TSK and T2 are lower than at the next higher eta levels, which underlines the well-known fact that near-surface temperature gradients are lower than in the free atmosphere (Rist et al., 2020). The free atmosphere is typically defined in WRF as the part of the atmosphere

above the planetary boundary layer, where turbulent mixing from the surface becomes negligible. However, the exact eta level or geopotential height that marks the transition to the free atmosphere is not fixed. It varies dynamically in space and time, depending on surface properties, atmospheric stability, and time of day (Hong et al., 2006).

From eta level 1 to eta level 20 (Fig. 2c and d), the linear gradient between WRF temperatures and surface elevation shows almost no scatter for the first eta levels. Beyond eta level 20, the scatter starts increasing, and there is no relationship anymore

at level 49. This can be explained by the fact that eta levels 1 to 20 follow the terrain, while the remaining eta levels up to 49 are pressure levels. The lapse rate at eta level 1 equals -0.79 °C 100 m$^{-1}$, i.e., it is higher than in the free atmosphere (which corresponds to -0.65 °C 100 m$^{-1}$) for the three-month test period and gradually decreases for higher levels.





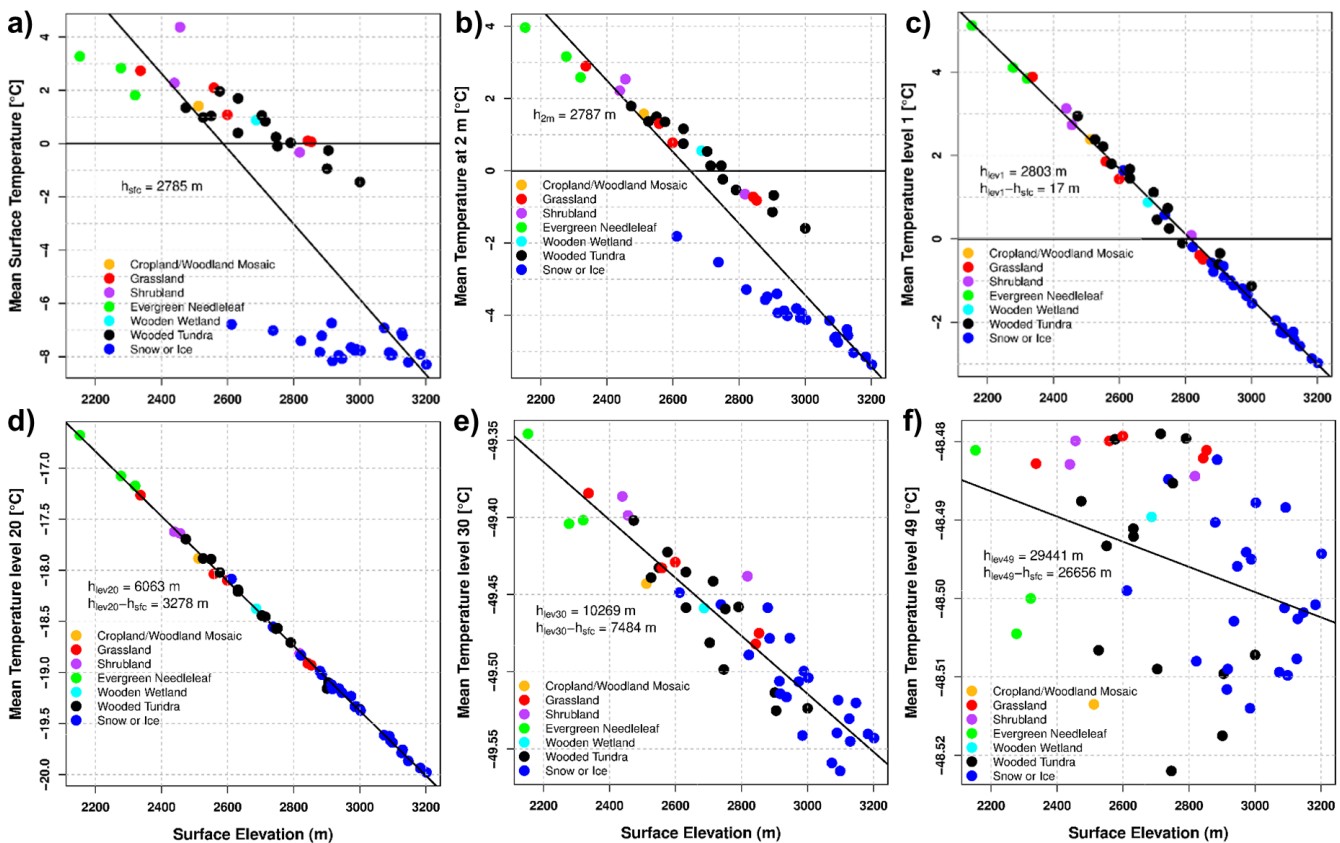

**Figure 2.** Mean WRF temperature for different land use categories and surface elevations (m) for the Kauner Valley domain covering a test period of three months. The diagonal black line shows the linear gradient between elevation and temperature obtained by least-squares fitting to all data points. The numbers in the sub-plots indicate the respective geopotential height and the height above the ground.

### 2.4.1 Correction of the WRF temperature

The adjustment of the WRF temperature is done in a three-step approach on hourly time series to consider sub-daily changes
in the temperature lapse rates, TLRs. First, we compute spatial averages of the hourly temperature of eta levels 1 and 20 over
the 200 km$^2$ domain of the upper Kauner Valley by applying Equation 1:

$$\bar{T}_{levelX,t} = \frac{1}{N} \sum_{i=1}^{M} \sum_{j=1}^{L} T(t,i,j) \tag{1}$$

where $\bar{T}_{levelX,t}$ is the spatial mean of temperature [°C] of the respective eta level ($levelX$) at time $t$. In our case, time $t$
has an hourly resolution. $\bar{T}_{levelX}(t,i,j)$ is the variable value at time $t$, latitude index $i$, and longitude index $j$. $M$ is the total
number of latitude points, and $L$ is the total number of longitude points. $N$ is the total number of spatial grid points. As the





WRF grid of the Kauner Valley has 7 rows ($M$) and 7 columns ($L$), a total of 49 grid cells ($N$) are considered in Equation 1. The spatial averaging makes the TLRs more robust regarding the spatial heterogeneity.

We use the same approach to calculate the spatial averages of the hourly geopotential height following Equation 2:

$$\bar{H}_{levelX,t} = \frac{1}{N} \sum_{i=1}^{M} \sum_{j=1}^{L} H(t,i,j) \tag{2}$$

where $\bar{H}_{levelX,t}$ is the spatial mean of geopotential height [m a.s.l.] of the respective eta level ($levelX$) at time $t$. As the temperature and geopotential height grids have the same spatiotemporal dimensions, the variables of $\bar{H}_{levelX}(t,i,j)$ are identical to $\bar{T}_{levelX}(t,i,j)$.

In a second step, we compute the hourly TLRs based on the spatial averages (from Equ.1 and 2) using Equation 3:

$$\tau_t = \frac{\bar{T}_{\text{level\_20},t} - \bar{T}_{\text{level\_1},t}}{\bar{H}_{\text{level\_20},t} - \bar{H}_{\text{level\_1},t}} \tag{3}$$

where $\tau_t$ [°$Cm^{-1}$] is the TLR at time step $t$, $\bar{T}_{\text{level\_20},t}$ [°C] is the spatial average temperature $\bar{T}$ of eta level 20 at time step $t$, and $\bar{H}$ is the spatial average altitude of the eta levels in m a.s.l. (i.e, the geopotential height). Eta level 20 corresponds to a geopotential height of about 6000 m a.s.l., which represents a mean height above ground of 3278 m. We have chosen the altitude range from eta level 1 to 20 for two reasons: eta level 1 still includes minor temperature differences due to the land-atmosphere interactions (Fig. 2c), which are more pronounced in summer (Fig. S2a). On the contrary, eta level 20 does

not show any influence of the surface properties (Fig. 2d) throughout the season (Fig. S2d). Accordingly, the higher variability and magnitudes from lower levels will be averaged out by the upper levels. Although the transition of mean TLR from eta level 1 to 20 is not linear, we apply the linear function of Equation 3 for computing mean TLR over the entire altitude range. The red line in Figure 3 represents the mean TLR between eta levels 1 and 20. Including eta level 1 in the linear equation ensures that temperature fluctuations are not completely decoupled from the surface and topography of the research area.

We compute the overall mean TLR using Equation 4:

$$\bar{\tau}_t = \frac{1}{Y} \sum_{t=1}^{Y} \tau(t) \tag{4}$$

where $\bar{\tau}_t$ is the mean TLR [°$Cm^{-1}$] over time and $Y$ is the total number of time steps. Accordingly, the computed mean TLR is -0.57 °C 100 m$^{-1}$ from eta level 1 to 20 over the period 1973-2015. The summer TLR (i.e., May to September) is with -0.58 °C 100 m$^{-1}$ slightly larger than the winter TLR (-0.56 °C 100 m$^{-1}$).

In the third step, we compute the hourly corrected WRF temperature ($T_{\text{cor}}$) [°C] at actual altitudes of meteorological stations by applying Equation 5:

$$T_{\text{cor,t}} = T_{\text{level\_1},t} + \tau_t * (H_{\text{station}} - H_{\text{level\_1},t}) \tag{5}$$

where $H_{\text{station}}$ represents the station elevation in m a.s.l. and $H_{\text{level\_1},t}$ the mean geopotential height of the eta level 1 (i.e., 2803 m a.s.l).





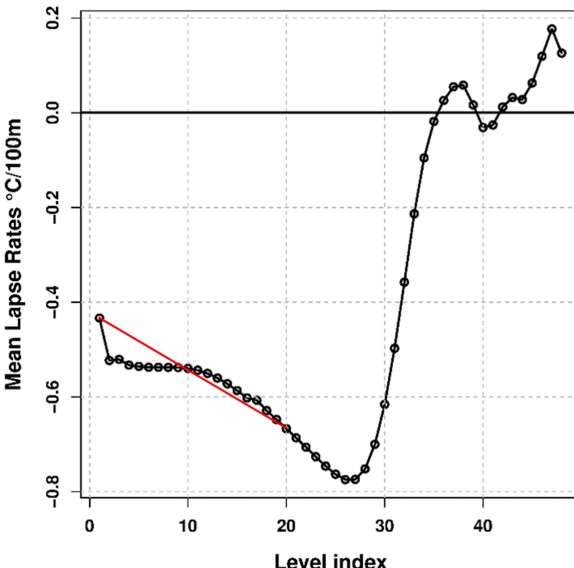

**Figure 3.** Mean WRF temperature lapse rates in °C (100 m)$^{-1}$ per eta level. The red line represents the mean lapse rate between eta levels 1 and 20.

To compare the uncorrected WRF temperature with the corrected WRF temperature ($T_{\text{cor}}$) and the observed station temperature, we account for the height difference between the WRF grid cell ($H_{\text{WRF}}$) and the actual station ($H_{station}$) by multiplying the height difference with the mean TLR of 5.7 °C km$^{-1}$ already calculated from Equation 4. The altitude adjusted uncorrected WRF temperature $T_{\text{uncor,alt,t}}$ is thus computed by applying Equation 6:

$$T_{\text{uncor,alt,t}} = T_{\text{uncor,t}} + (H_{\text{WRF}} - H_{\text{station}})0.0057 \tag{6}$$

where $T_{\text{uncor,t}}$ is the hourly uncorrected 2 m temperature of a WRF cell, $H_{\text{WRF}}$ represents the WRF altitude, and $H_{station}$ the actual station altitude in m a.s.l..

### 2.4.2 Correction of the WRF precipitation

Climate models, including the WRF model, often overestimate the frequency of light rainy days while simultaneously underestimating the total amounts of extreme observed precipitation (Lazoglou et al., 2024). The model's tendency to overestimate the occurrence of light precipitation events is known as the "drizzle bias" (Lazoglou et al., 2024; Velasquez et al., 2020). We follow the approach of Lazoglou et al. (2024) to adjust the number of rainy days based on the assumption that the relationship between observed and simulated rainy days remains the same in time (thresholding). The correction of the dry-days frequency with thresholding was already applied in several studies before (e.g., Smitha et al., 2018; Velasquez et al., 2020). Thresholding is a widely used, simple, and efficient method for correcting the frequency of wet days. It involves converting all simulated values below a designated threshold to zero (Lazoglou et al., 2024). We adjust the WRF precipitation frequency by computing



**Table 2.** Monthly mean and standard deviation of observed precipitation days at station Vergoetschen and derived threshold values for correcting the WRF precipitation frequency from 1971 to 2015.

| Month | Mean prec days | SD prec days | Threshold (mm h$^{-1}$) |
|:-----:|:--------------:|:------------:|:-----------------------:|
| 1 | 9 | 4.83 | 0.41 |
| 2 | 8 | 3.81 | 0.49 |
| 3 | 10 | 4.06 | 0.42 |
| 4 | 11 | 3.93 | 0.48 |
| 5 | 13 | 4.37 | 0.43 |
| 6 | 16 | 4.52 | 0.42 |
| 7 | 15 | 3.6 | 0.29 |
| 8 | 15 | 4.58 | 0.27 |
| 9 | 12 | 3.96 | 0.23 |
| 10 | 10 | 4.43 | 0.29 |
| 11 | 10 | 4.17 | 0.33 |
| 12 | 10 | 4.32 | 0.41 |

wet-day frequencies in both monthly observed and simulated datasets. As the "drizzle bias" has a distinct seasonality (see Fig. S3) and is consequently more pronounced in the winter period, we computed mean monthly thresholds based on the observed and simulated counts of rainy days. The first step is to compute the mean monthly precipitation days based on daily station data. We select the meteorological station Vergoetschen (1280 m a.s.l.), which HD-Tyrol operates, as the station has a long record

of daily precipitation starting in 1971 without data gaps. Observed precipitation days are defined as days with precipitation exceeding 0.01 mm d$^{-1}$. Observed mean monthly precipitation days from 1971 to 2015 are listed in the second column of Table 2.

We estimate monthly-varying precipitation thresholds to identify WRF time steps whose precipitation is to be set to zero for frequency matching. These thresholds are identified based on the absolute minimum error between observed and WRF

precipitation days for hourly resolution (Tab. 2). Figure S4 of the supplementary material shows a graphical illustration of the optimization approach. In the second step, the hourly WRF precipitation values below the monthly threshold values (Tab. 2) are set to 0, assuming the monthly-varying precipitation thresholds are stationary over the period 1971 to 2015, space, and elevation. The correction of WRF precipitation only leads to a reduction of the annual precipitation over the research site by 4 % compared to the uncorrected, as mainly low precipitation frequencies are set to 0. WRF precipitation intensities (i.e.,

quantiles) are not evaluated or adjusted due to a lack of reliable precipitation observations from higher elevations (> 2000 m a.s.l.).





## 2.5 Statistical tests

We use different statistical tests to evaluate the properties of observed and modeled time series. We test the significance of trends with the non-parametric and modified Mann-Kendall test for serially correlated data using the Hamed and Rao (1998) variance

correction approach, which is available in the "*modifiedmk*" R package version 1.6 (URL in the *Code and data availability* section). This approach reduces significant serial dependency (i.e., autocorrelation) in time series by calculating the effective sample size. The effective sample size is calculated using the ranks of significant serial correlation coefficients, which are then used to correct the inflated (or deflated) variance of the test statistic (Hamed and Rao, 1998).

We apply the non-parametric Kolmogorov–Smirnov (KS) test to evaluate the similarity between empirical cumulative dis-

tribution functions (ECDFs). The KS test compares the two ECDFs by quantifying the maximum distance between them. We use the *ks.test* function (version 1.5.1) in R (URL in the *Code and data availability* section). Besides the p-value, the output of the KS test includes the D-statistic, which is the dimensionless maximum absolute difference between ECDFs. In addition, we compute the maximum vertical difference between the ECDFs.

The non-parametric Pettitt test detects a sudden change in the central tendency (i.e., mean) of the time series (Pettitt, 1979).

The H0-hypothesis, no change, is tested against the HA-hypothesis, change. The applied Pettitt test is available in the "*trend*" R package version 1.1.6 (URL in the *Code and data availability* section).

## 2.6 WaSiM model setup

The grid-based Water Flow and Balance Simulation Model (WaSiM) was initially developed for climate and land use change studies in mountainous regions (Schulla, 2021). In previous studies, WaSiM has proven to be a powerful tool for investigating

the spatial and temporal variability of hydrological processes in complex river basins (e.g., Strasser et al., 2019). WaSiM uses physics-based modeling approaches for simulating the vertical water flux in the unsaturated zone based on the Richards equation and the energy-balance approach for simulating snow melt. Additionally, WaSiM can consider snow redistribution processes induced by wind or gravitation, which is highly relevant for high-elevation alpine catchments (Warscher et al., 2013). A detailed explanation of the snow routine can be found in the WaSiM documentation and the publication of Hofmeister et al.

(2022). The model parameters for simulating gravitational slides (see Table 3) are identical to those of Hofmeister et al. (2022) as the spatial and temporal resolution of the model setup is the same (i.e., hourly time step and 25 m grid resolution). The range of the main wind direction for wind-induced snow redistribution was derived from monthly mean WRF data for the winter period (from October to May). Accordingly, WaSiM corrects snowfall considering the exposition, i.e., snowfall from wind-exposed grid cells is moved to sheltered cells. Consequently, the algorithms for simulating wind-induced snow redistribution

and gravitational slides are not event-based, as they try to match snow accumulation at the end of the winter. The temperature threshold (T0R) is set to 2 °C, which is in agreement with other studies conducted in the Alpine region (e.g., Frei et al., 2018). The temperature transition range from snow to rain is set to 2 °C (i.e., half snowfall and half rain), allowing a smooth transition between rain and snowfall from 0 °C (i.e., only snowfall) to 4 °C (i.e., only rain). The correction factors for the incoming and outgoing long-wave radiation are set to 1, meaning no correction is applied.





A variety of algorithms are available in WaSiM for the interpolation of meteorological inputs. Most of the five input vari-
ables (i.e., relative humidity, precipitation, wind speed, and short-wave incoming radiation) are interpolated with a bilinear
approach from the 2 km WRF to the 25 m grid resolution of WaSiM. Temperature is inter- and extrapolated based on an
elevation-dependent regression approach. The interpolation algorithms are applied on an hourly time step, which allows the
consideration of variable temperature lapse rates, which is far more realistic than a fixed temperature lapse rate. Short-wave in-

coming radiation and temperature are adjusted according to topographic shading following the scheme devised by Oke (2002).
We assume a positive precipitation lapse rate of 0.5 % per 100 m, which is a common assumption for the Ötztal Alps (Schöber
et al., 2014; Schmidt et al., 2022), for re-scaling the relatively coarse WRF precipitation from the 2 km grid resolution to the
actual topography of the much finer grid resolution of WaSiM (i.e., 25 m). With the activated snow redistribution module of
WaSiM, preferential snow erosion and deposition are considered and simulated.

Glacier melt and evolution are particularly important for hydrological studies in glaciated catchments such as the upper
Kauner Valley. For this, WaSiM offers a dynamical glacier module that adds or removes glaciated grid cells dependent on the
specific mass balance (Stahl et al., 2008). The standard threshold value for transforming a grid cell into a glacier cell is 2000
mm SWE. However, WaSiM does not consider the actual ice flow of a glacier. Hence, positive mass gains are not transported
from the accumulation to the ablation area of a glacier, which has certain limitations in areas with high glacier dynamics. A

recently published coupling scheme of WaSiM and the Open Global Glacier Model (OGGM) bridges the gap between more
sophisticated glacier evolution models and local catchment hydrology (Pesci et al., 2023). For initialization, WaSiM estimates
the volume of a glacier based on the volume-area scaling (VAS), which offers the main advantage of minimum input data,
as glacier extent data are usually available even in data-scarce regions. According to Bahr et al. (2015), the dimensionless
exponent of the VAS equation is fixed to 1.375, while the scale factor depends on the sample data set. Since the scale factor can

only be estimated for larger sample sizes of around 50 or more glaciers to ensure statistical robustness, an already published
scaling factor for the European Alps from Chen and Ohmura (1990) is chosen. Additionally, the dynamic glacier routine of
WaSiM enables the modeling of the metamorphism of snow to firn after one balancing period and to ice after six additional
balancing periods (hydrological years). The initialization of the firn layer is based on the WEchnge parameter that defines
the rate of change in the water equivalent (WE) of firn with each meter of elevation increase (default is 1.8 mm m$^{-1}$) and

the average equilibrium line (ELA), which separates the accumulation and ablation area of a glacier. As no continuous ELA
information is available for the glaciers of the upper Kauner Valley, we use the ELA in m a.s.l. from a neighboring glacier,
the Vernagtferner (WGMS, 2025). Glacier melt is simulated by an extended degree-day approach initially developed by Hock
(1999), which includes information on global radiation during each time step on each grid cell to modify the ice and firn
melt. Besides a classical melt factor (MF), one empirical coefficient for minimum radiation and one for maximum radiation

are needed. Hence, glacier melt increases when shortwave radiation input is high and vice versa. Snow melt on the glacier is
simulated by the same energy-balance approach as for non-glaciated grid cells to ensure consistency.

In general, we used the standard model parameters from the WaSiM documentation whenever possible (Tab. 3), as the
main focus of this study is to investigate the sensitivity of WaSiM with respect to changed forcing data. Although calibration
of the main WaSiM parameters can improve model results, we have refrained from parameter optimization to investigate



**Table 3.** Key processes and main WaSiM parameters.

| Process | WaSiM Parameter | Description | Value |
|---|---|---|---|
| Snow accumulation | T0R | Temperature limit for rain (°C) | 2 |
| | Ttrans | Temperature-transition range from snow to rain (°C) | 2 |
| Gravitational redistribution | $i_{lim}$ | Maximum deposition slope (°) | 55 |
| | $D_{lim}$ | Scaling for maximum deposition (mm) | 2 |
| | $i_{erosion}$ | Minimum slope for creating slides (°) | 50 |
| | $f_{erosion}$ | Fraction of snow pack that forms the slide (-) | 0.002 |
| Snow ablation | LWINcorr | Correction factor for incoming long-wave radiation (-) | 1.0 |
| | LWOUTcorr | Correction factor for outgoing long-wave radiation (-) | 1.0 |
| Wind redistribution | start azimuth | 1st quantile of wind direction (°) | 179 |
| | end azimuth | 3rd quantile of wind direction (°) | 217 |
| | cmin | Minimum correction factor (-) | 0.5 |
| Glacier initialization | VAscale | Scale factor (-) | 28.5 |
| | VAexp | Exponent (-) | 1.375 |
| | WEchnge | Change rate of WE per m (mm) | 1.8 |
| | ELA | Average equilibrium line elevation (m) | 3113 (1969); 3200 (2006) |
| Glacier melt | MF | Melt factor (mm) | 2 |
| | $ice_{min}$ | Minimum radiation coefficient for ice (mm Wh$^{-1}$ m$^2$ °C$^{-1}$ day$^{-1}$) | 0.0001 |
| | $ice_{max}$ | Maximum radiation coefficient for ice (mm Wh$^{-1}$ m$^2$ °C$^{-1}$ day$^{-1}$) | 0.0007 |

the propagation of the changed forcing to the different model results. This means all model parameters are identical for all
simulation runs. We use WaSiM version 10.07.02 for all simulations in this study.

## 3 Results

### 3.1 WRF temperature at station locations

As a first evaluation, we compare the WRF temperature with station observations from the upper Kauner Valley and surrounding
(Fig. 1 and Table 1). The uncorrected WRF temperature corresponds to the WRF hourly temperature of the respective 2 km grid
cell, meaning it represents the simulated near-surface (i.e., 2 m) temperature and is adjusted for the height difference between
WRF and station altitude by applying Equation 6. The corrected WRF temperature corresponds to the hourly temperature
computed from the WRF atmosphere applying Equation 5. We compare the general statistics of the time series in the form of





ECDF, as plotted in Figure 4. The periods of the ECDFs always refer to all data since the start of measurement recording (Tab.

1) until 31 December 2015. No data values (NAs) in the observations are set to no data in the WRF temperature time series. The lowest station is Dammfuss, at an elevation of 1770 m a.s.l., with continuous temperature records since 1988, followed by the Gepatschalm station, which was installed at an altitude of 1900 m a.s.l. in 2009. The second highest station is Weisssee, at an elevation of 2540 m a.s.l., with continuous temperature observations since December 2006. The station Vernagt Pegel was installed at 2640 m a.s.l. in 1975. Consequently, the selected stations cover an elevation difference of almost 900 m.

The four panels of Figure 4 compare the ECDF of station temperature (black line), of the uncorrected WRF temperature (red line), and the corrected WRF temperature (blue line). We evaluate the similarity between the different ECDFs with the KS test. The results of the KS tests are listed in Table S1 of the supplementary material. At Dammfuss, the uncorrected near-surface WRF temperature agrees with the frequency of observed mean and warmer (> 5 °C) temperatures but shows a positive bias (higher frequency) for colder temperatures (see Fig. 4a). The frequency distribution of the corrected WRF temperature aligns

with the observations at Dammfuss station and only has a slight cold bias for temperatures < 0 °C. According to the computed D-statistics (Tab. S1), the maximum distance between the observed and WRF temperature ECDFs is 0.08 for the uncorrected and 0.03 for the corrected temperature. The maximum vertical difference between the ECDFs is at -3.06 °C in the case of uncorrected and at 17.46 °C for the corrected.

At Gepatschalm station, the critical temperature frequencies are more towards warmer temperatures (> 5 °C) (see Fig. 4b).

The correction of the WRF temperature reduces the warm bias in the range from 5 °C to 15 °C, but there is still a slight offset. Conversely, the cold bias in the near-surface WRF temperature is absent in the corrected time series. The maximum distance between the ECDFs is 0.08 in the case of the uncorrected and 0.07 for the corrected WRF temperature (Tab. S1).

We can also observe a similar cold bias (< 0 °C) between the observation and uncorrected WRF temperature at Weisssee station (see Fig. 4c). Using the WRF atmosphere, the cold temperature offset can be corrected, which results in a good agreement

with the observed temperature records. The maximum distance between the ECDFs is 0.05 in the case of the uncorrected and 0.03 for the corrected WRF temperature (Tab. S1).

The largest cold bias covering the entire temperature range is present at the highest station, the Vernagt Pegel (see Fig. 4d). The corrected WRF temperature shows a more realistic representation of temperature frequencies at 2640 m a.s.l. altitude. The respective land cover class of the WRF grid cell is glacierized. Therefore, the maximum distance between the ECDFs is much

larger (i.e., D-statistics uncorrected 0.13 and corrected 0.03) compared to the lower-elevation stations (Tab. S1).

Although the comparison with the station data only represents specific elevation ranges and locations of the research area, it indicates that the near-surface WRF temperature has difficulties representing the small-scale heterogeneity in surface-albedo characteristics, especially for the higher altitudes (> 2600 m a.s.l.). Hence, the correction approach based on temperature lapse rates from the WRF atmosphere provides more realistic temperature predictions independent of the surface-albedo character-

istics.

Figure 5 compares seasonal hourly observed and corrected WRF temperatures. The comparison is done for different data subsets depending on the availability of hourly temperature records. The seasonal temperature dynamic is well reproduced by the corrected WRF temperature at all observation sites. When comparing the median of the four boxplots, we observe a slight



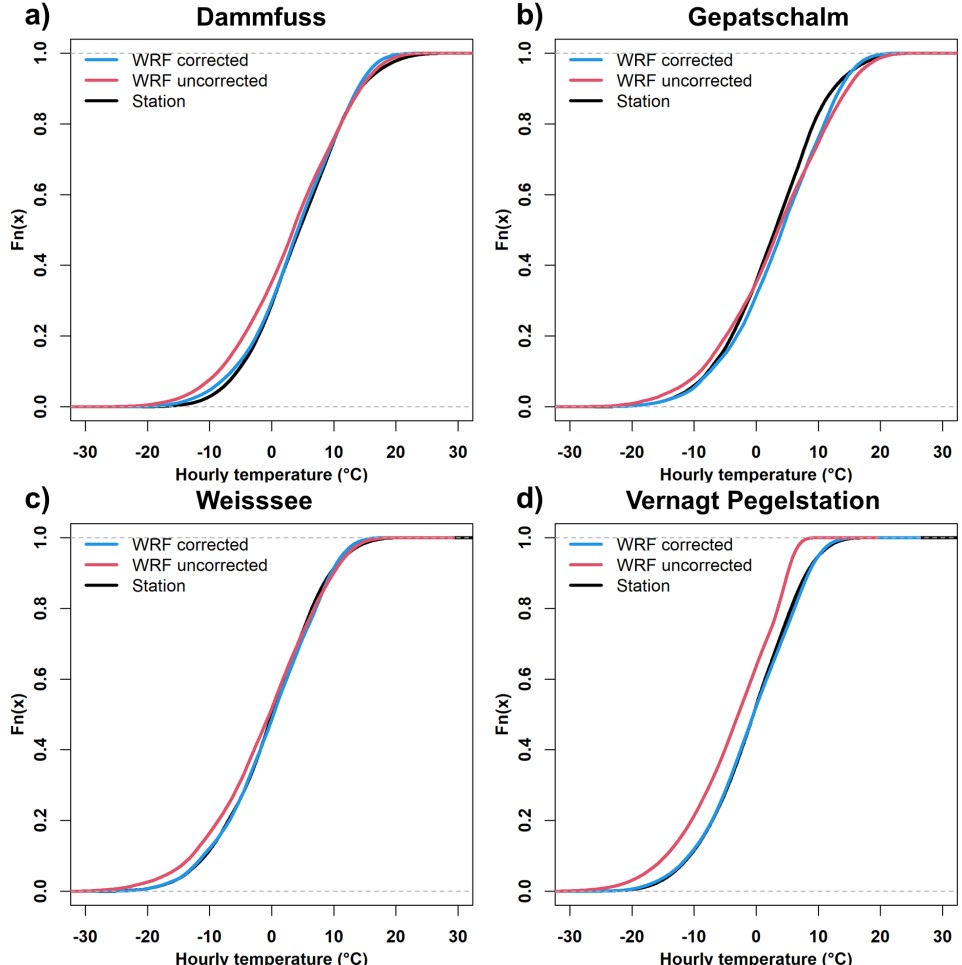

**Figure 4.** Empirical cumulated distribution functions (ECDF) of observed hourly temperature (black line), uncorrected WRF temperature (red line), and corrected WRF temperature (blue line) at different stations a) Dammfuss, b) Gepatschalm, c) Weisssee, and d) Vernagt Pegelstation.

seasonal tendency of the corrected WRF to overestimate summer temperature. In addition, the standard deviation is under-
or overestimated by the corrected WRF temperature for some months. Nevertheless, we cannot observe a systematic false
behavior of the corrected WRF temperatures at the different sites and altitudes.




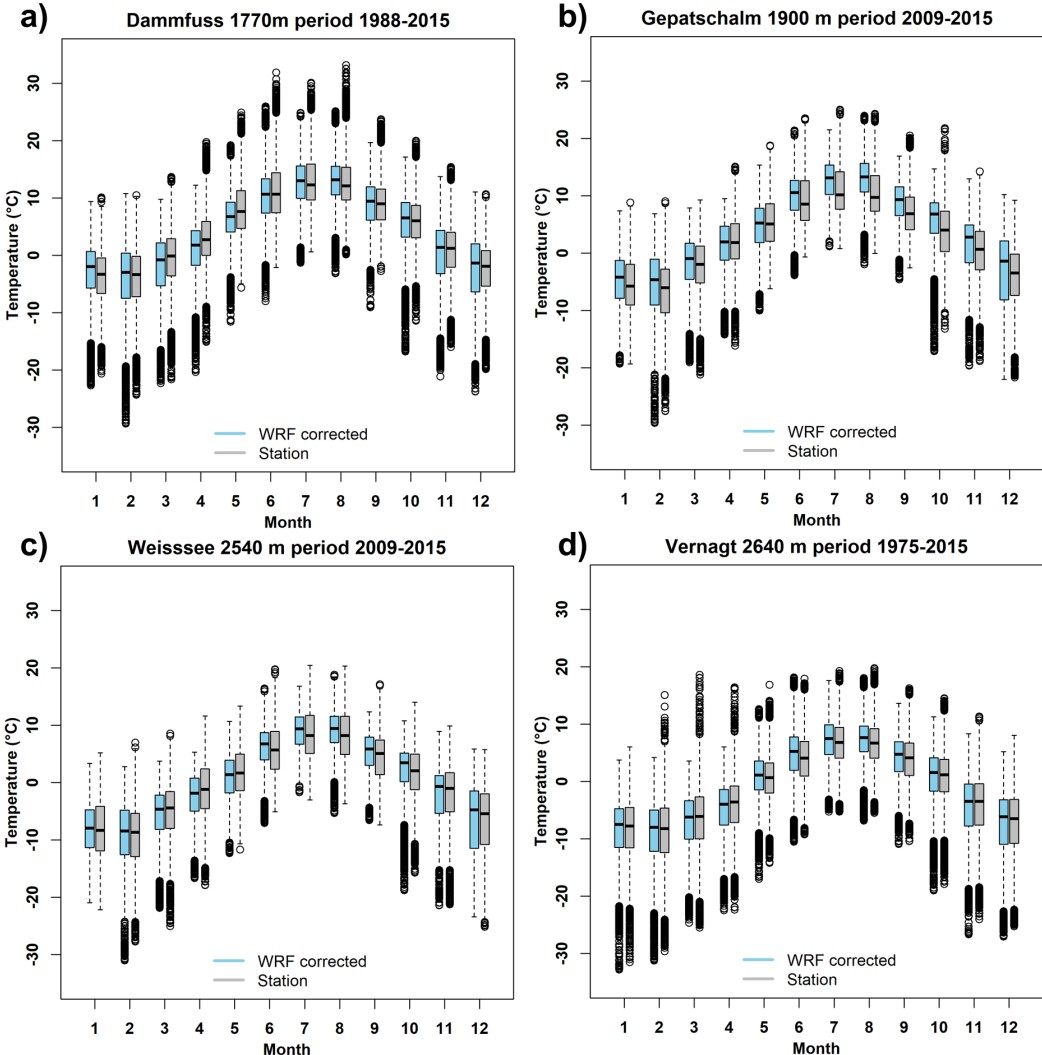

**Figure 5.** Monthly boxplots of hourly observed temperature (grey) and corrected WRF temperature (blue) at different stations a) Dammfuss, b) Gepatschalm, c) Weisssee, and d) Vernagt Pegelstation. The different temporal coverages are given in the titles of the respective subplots.

## 3.2 Robustness of temperature trends and elevation-dependent warming

In order to evaluate the robustness of the WRF temperature trend, we compare it with a long-term and homogenized observation from the HISTALP data set (Auer et al., 2007). Monthly mean temperature records since 1851 are available at the Obergurgl

station, which is approximately 20 km away from the Kauner Valley and located at 1936 m a.s.l.. As this particular site is located in a neighboring valley (i.e., the Ötz Valley), it is also a test of the general transferability of the applied correction method for the WRF temperatures. Figure 6 shows the evolution of monthly and annual temperatures at the station Obergurgl. In general, the WRF temperature follows well the evolution of the observed monthly and annual temperature. The mean



temperature at Obergurgl station is 2.16 °C in the observation from 1973 to 2015. The mean of corrected WRF temperature
at the respective elevation (1936 m a.s.l.) is 2.61 °C. Consequently, WRF overestimates the mean temperature by 0.45 °C.
The mean annual temperature increase is 0.06 °C at Obergurgl station (Sen's slope of 0.003 °C a$^{-1}$) and 0.05 °C for the
corrected WRF temperature (Sen's slope of 0.003 °C a$^{-1}$). Both temperature increases represent a significant positive trend
with a confidence interval of 0.95. The Pettitt test detects a probable change point, meaning a sudden change in the central
tendency (i.e., mean), in both time series for May 1987. The sudden warming in the European Alps at the end of the 1980s was
already reported by several other studies (e.g., Marty, 2008; Marcolini et al., 2019; Hofmeister et al., 2023). Coming back to
our third research question on the consistency of the bias-corrected RCM data over a longer period (> 30 years) with respect
to non-stationarity conditions: our results clearly underline this consistency, given that the statistical properties are well in line
with the observations.

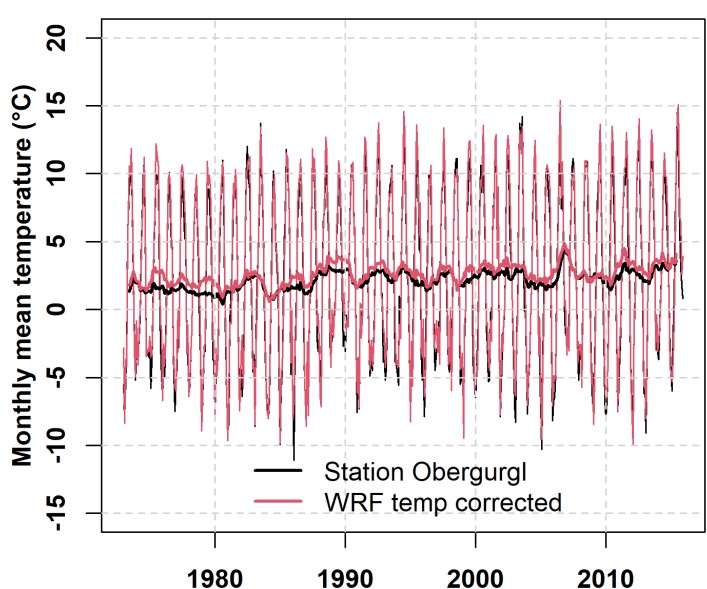

**Figure 6.** Mean monthly temperature at Obergurgl station (thin black line) and corrected WRF temperature (thin red line) at the respective
station altitude. The thicker lines illustrate the respective yearly moving average.

In a further analysis, we test whether the temperature increase has different trends concerning the elevation of the research
area. We first test the significance of the temperature trends with the modified Mann-Kendall test and second, compute the coef-
ficient of a linear model giving the annual temperature at four different altitudes (Tab. 4). The first three elevations correspond
to actual meteorological observation stations in Kauner Valley, while the highest (i.e., 3800 m a.s.l.) represents the highest
point of the research area. Interestingly, we find stronger warming by 17 % when comparing the lowest and highest locations
of the research area, meaning the higher altitudes are experiencing a stronger temperature increase. This observation is also in





**Table 4.** Statistics on temperature evolution with respect to elevation, p-value of modified Mann-Kendall test, Sen's slope, and mean annual temperature trend from 1973 to 2015

| Elevation (m a.s.l.) | mMK p-value | Sen's slope (°C a$^{-1}$) | Trend (°C a$^{-1}$) |
|---|---|---|---|
| 1770 | 0.0010 | 0.0031 | 0.0356 |
| 1900 | 0.0008 | 0.0031 | 0.0361 |
| 2540 | 0.0004 | 0.0032 | 0.0384 |
| 3800 | 0.0001 | 0.0035 | 0.0429 |

line with other studies on the aspect of elevation-dependent warming in the European Alps (e.g., Rangwala and Miller, 2012; Kuhn and Olefs, 2020).

### 3.3 Impact of temperature correction on hydrological model results

We use the HM WaSiM to investigate the impact of the WRF temperature correction on the catchment scale and for different hydrological processes, such as snow, glacier, and streamflow generation. The simulation period is from 1 October 2006 until 410 30 September 2015, but the actual period for the analysis is a bit shorter (eight years) as the first simulation year is considered as a spin-up period. The multi-data evaluation of the simulation results is limited by the temporal coverage of the WRF data, which is only available until 31 December 2015 in this study. Hence, the overlap of observed snow depth and snow cover data is relatively short. Nevertheless, the simulation period is long enough to observe similarities and discrepancies between corrected and uncorrected model runs.

In Figure 7a, we compare the ECDFs of daily observed SWE at Weisssee station (black line) with simulated SWE at the respective 25 m grid cell of WaSiM. In general, WaSiM tends to overestimate SWE with both corrected and uncorrected model forcing. The simulated SWE seems to be relatively insensitive to the WRF temperature inputs at this particular grid cell. The mean daily SWE is 163 mm in the observation, while the simulated SWE with corrected temperature is 187 mm and 195 mm in the case of uncorrected temperature. Although the comparison of in situ SWE with simulated SWE comes with several 420 assumptions and uncertainties due to the high spatial heterogeneity of snow processes, the similar behavior of the simulated SWE gives an interesting insight into snow accumulation at 2540 m a.s.l. altitude.

Figure 7b shows the ECDFs of daily observed and simulated fractional snow-covered area (fsca). Daily observed snow cover data are from MODIS (black line). Not surprisingly, the agreement between observed and simulated fsca is largest for the winter period, in which the entire research area is covered by snow (Fig. S5). However, the discrepancy between observation 425 and simulation, but also between the two temperature forcing increases in the period of low fsca. Especially, the simulated fsca with uncorrected WRF temperature input overestimates snow cover in the research area from the range < 0.6 fsca. The ECDF of simulated fsca with corrected WRF temperature is more in line with the fsca of MODIS, especially at low to mid fsca values (i.e., fsca < 0.6). The negative snow cover bias in the snow-free period (fsca < 0.3) can have several reasons, such as the heterogeneity of the snow pack, the limited sensor resolution of MODIS, and the processing of the fsca. Nevertheless, the





fsca data cover the entire research area, in contrast to the in situ SWE data, and provide valuable information on the seasonal evolution of the snow cover.

The ECDFs of hourly streamflow at the gauge Gepatschalm are illustrated in Figure 7c. The observed streamflow (black line) shows a typical behavior for a snow and glacier-dominated catchment with an extended low flow period with a median of 0.85 m$^3$ s$^{-1}$. The simulated streamflow with uncorrected WRF temperature significantly underestimates all parts of the
observed ECDF, meaning WaSiM generates too little streamflow in all seasons (Fig. S6). The WaSiM run with corrected WRF temperature shows a more realistic behavior and follows the ECDF of the observed streamflow. Although the mean annual streamflow is well estimated with the corrected WRF temperature (i.e., the mean flow of observation and simulation are both 3.1 m$^3$ s$^{-1}$), WaSiM overestimates the higher streamflow and underestimates the lower streamflow. Hence, there is potential for optimizing the seasonal streamflow generation of WaSiM (Fig. S6b). We compute the goodness-of-fit criteria for comparing
the daily observed and simulated streamflow for the respective period. Although WaSiM provides hourly streamflow outputs, we compare daily values as they are expected to be less impacted by the fact that WRF does not assimilate any precipitation station data from that particular region, which limits the direct comparison on an hourly resolution. In the case of the daily values, we assume that the model performance depends more on the corrected prediction of snow and glacier dynamics than on intense precipitation events. The daily goodness of fit criteria with uncorrected WRF temperature are KGE 0.42, percent
bias -46, and RMSE 2.5 m$^3$ s$^{-1}$. While the daily performance with corrected WRF temperature is much better with KGE 0.71, percent bias -0.7, and RMSE 2.4 m$^3$ s$^{-1}$. Note that the only difference between the WaSiM simulations is the WRF temperature forcing. We did not apply a calibration strategy (i.e., manual or auto calibration) to optimize the model parameters according to the model forcing. This shows how important the correct meteorological driving forces are for the model parameterization, also in physics-based models, in order to achieve high scores in the goodness-of-fit criteria.

The simulated cumulated glacier mass balances indicate the highest sensitivity concerning the temperature forcing (see Fig. 7d). The cold bias of the near-surface WRF temperature (red line) leads to a constant positive glacier mass balance, which is quite unrealistic for the period 2008 to 2015. In the highest parts of the research area (> 2700 m), the cold bias results in too low snow melt and, hence, an overestimation of the snow cover. As long as the glaciers are snow-covered and not exposed to the incoming radiation, firn and ice melt do not occur on the glacier. When forcing WaSiM with the corrected WRF temperature,
the simulated glacier mass balances show a stronger seasonality and a continuous negative trend, which would be expected for this region. The only exception in the negative trend is the winter period of 2013/14, which was particularly snowy, and the following summer was cooler and wetter. The mass balance year 2013/14 was almost balanced in the Central European Alps with only a slightly negative average balance of -174 mm (WGMS, 2017). At the end of the simulation period, we observe a deficit of almost 6000 mm in the water balance of the catchment. As snow and glacier melt are the dominant streamflow
components in the upper Kauner Valley in this particular period, the deficit of meltwater directly propagates to the streamflow generation.

Table 5 summarizes the main water balance components simulated with the corrected and uncorrected WRF temperatures. Since the only difference in the model forcing is the temperature input, the total precipitation is identical between the two model runs. The mean annual precipitation sum is 1487 mm for the period 2008-2015. Due to the cold bias in the uncorrected WRF





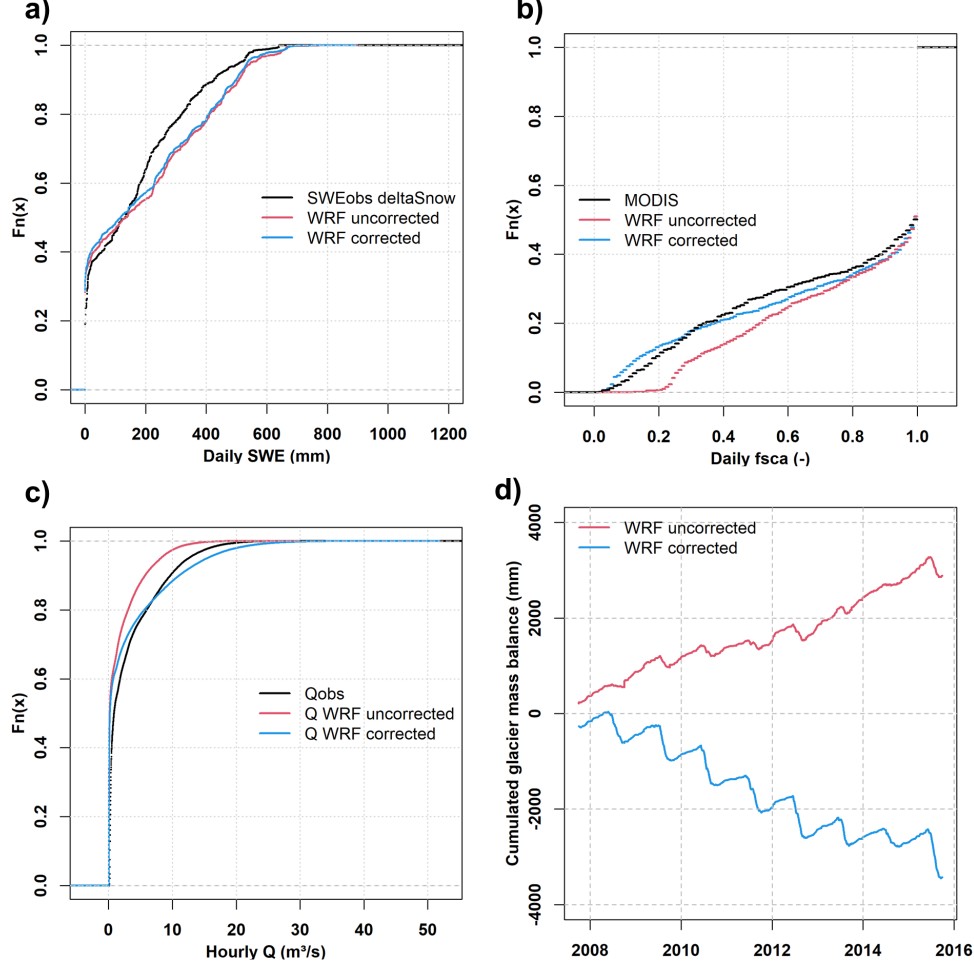

**Figure 7.** Empirical cumulated distribution functions of observed (black line) and simulated a) SWE at station Weisssee, b) fractional snow covered area (fsca), c) streamflow at gauge Gepatschalm, as well as d) simulated glacier mass balances averaged over the catchment. Simulation results with uncorrected WRF temperature (red line) and corrected WRF temperature (blue line).

temperature, the mean snowfall fraction of total precipitation is slightly higher (68 %) than that of the corrected simulation run (62 %). Consequently, the fraction of rain in precipitation is smaller for the simulation run with uncorrected WRF temperature. Despite the inter-annual variability of streamflow magnitudes, the simulated mean annual streamflow sum matches quite well with the observation as it is underestimated only by < 1 %. The simulated streamflow with the uncorrected WRF temperature underestimates annual streamflow by almost 50 % (Fig. S6a). The mean annual evapotranspiration (ETR) is about 9 % smaller

in the case of uncorrected WRF temperature. Based on these facts, we can assume that the critical aspect is not the snowfall or ETR amount but rather an issue with the timing and magnitude of simulated snow melt that requires positive temperatures, especially in higher-elevation parts of the catchment. While covered with snow, WaSiM does not compute firn and ice melt



**Table 5.** Water balance components from simulation results without (uncor) and with (cor) WRF temperature correction in mm. Obs stands for observation, ETR for evapotranspiration, and GLMB for glacier mass balance.

| Component | 2008 | 2009 | 2010 | 2011 | 2012 | 2013 | 2014 | 2015 | Mean |
|---|---|---|---|---|---|---|---|---|---|
| Prec cor | 1589 | 1398.6 | 1353.86 | 1166.07 | 1466.79 | 1813.02 | 1488.37 | 1624.12 | 1487.48 |
| Prec uncor | 1589 | 1398.6 | 1353.86 | 1166.07 | 1466.79 | 1813.02 | 1488.37 | 1624.12 | 1487.48 |
| Snow cor | 998.8 | 966.09 | 816.99 | 585.3 | 962.9 | 1204.76 | 909.02 | 983.18 | 928.38 |
| Snow uncor | 1088.52 | 1022.72 | 884.04 | 676.34 | 1024.41 | 1291.66 | 989.67 | 1076.11 | 1006.68 |
| Rain cor | 493.31 | 333.42 | 457.65 | 518.93 | 403.36 | 485.78 | 487.81 | 542.94 | 465.4 |
| Rain uncor | 386.6 | 265.94 | 377.95 | 412.4 | 329.99 | 384.22 | 392.98 | 434.61 | 373.09 |
| Q obs | 1798.11 | 1769.95 | 1613.95 | 2226.59 | 1984.5 | 1627.43 | 1598.45 | 1754.66 | 1796.71 |
| Q cor | 1807.1 | 1659.78 | 1856.06 | 1657 | 1904.33 | 1777.37 | 1407.46 | 2230.33 | 1787.43 |
| Q uncor | 900.64 | 876.64 | 1009.37 | 879.13 | 1053.32 | 959.91 | 795.66 | 1294.54 | 971.15 |
| ETR cor | 65.24 | 83.23 | 55.65 | 102.81 | 75.81 | 43.16 | 44.25 | 71.26 | 67.68 |
| ETR uncor | 54.82 | 72.62 | 54.04 | 84.47 | 74.73 | 39.16 | 44.69 | 68.42 | 61.62 |
| GLMB cor | -396 | -419.05 | -594.76 | -638.63 | -628.03 | -161.18 | -54.28 | -737.19 | -453.64 |
| GLMB uncor | 389.17 | 449.13 | 269.4 | 153.11 | 213.83 | 665.22 | 590.35 | 199.17 | 366.17 |

on glaciated cells, which leads to a significant discrepancy in the simulated glacier mass balances, which is clearly visible in Figure 7d.

## 3.4 Longer simulation run (1973-2015)

With the relatively short simulation period of eight years, it is shown that the near-surface WRF temperature does not lead to plausible HM results. On the contrary, using the TLRs from the WRF atmosphere results in reliable model predictions for several processes, such as snow and glacier melt and streamflow generation in general. Here we test how robust and consistent the HM results are with the corrected WRF forcing over a longer period of 42 years. We initialize the model in 1969 with glacier extents from 1968 and consider a spin-up period of three years, meaning the actual analysis period began in 1973. As almost continuous observed streamflow data are available from 1971 onwards at gauge Gepatschalm, we compare the hourly and daily streamflow over different periods (see Fig. 8). The left panels of Figure 8 illustrate the mean and standard deviation of the streamflow in days of the year. The subplots in Figure 8 (a to e) present observed and simulated streamflow for shorter periods of 14 years. Figure 8a shows the earliest period from 1973 to 1987. The blue line of the mean simulated streamflow follows well the seasonality and magnitude of the mean observed streamflow (black line). However, when comparing the standard deviation of streamflow, WaSiM overestimates the variability of streamflow in spring and summer. The mean standard deviation observed from 1973 to 1987 is 1.2 $m^3$ $s^{-1}$ and that of the simulation is 1.6 $m^3$ $s^{-1}$. In general, the annual streamflow regime has a typical pattern of a nival regime. The ECDFs of hourly streamflow (Fig. 8b) indicate a good fit in the mean and high flow parts. However, WaSiM tends to underestimate low flows. The median of hourly streamflow is 0.8 $m^3$ $s^{-1}$ for the





observation and 0.2 m³ s⁻¹ for the simulation. The mean of hourly streamflow is 2.8 m³ s⁻¹ for the observation and 2.5 m³ s⁻¹ for the simulation.

The period 1987 to 2001 of snapshot Figure 8c is characterized by slightly higher streamflow magnitudes with increased daily amplitudes. The mean standard deviation of observed streamflow increases to 1.3 m³ s⁻¹ and to 1.7 m³ s⁻¹ in the simulation. The mean observed streamflow is 2.8 m³ s⁻¹ and the simulated 2.7 m³ s⁻¹. In general, the summer streamflow period is extending

into autumn.

In the following period (2001-2015), the high flow period further extends in the summer months (see Fig. 8e). Additionally, the streamflow generation shifts towards spring. As a consequence, the mean observed (3.1 m³ s⁻¹) and simulated (3 m³ s⁻¹) streamflow increases. The reason for the increased streamflow is the increased temperature (i.e., mean annual temperature is -3.5 °C in 1973-1987 and -2 °C in 2001-2015), which leads to a higher glacier melt contribution (Fig. 8b). Although WaSiM

well predicts the observed seasonality, it overestimates the mean streamflow in August and September.

The daily mean of streamflow values computed over the entire simulation period gives a good overview of the general tendency of the WaSiM model for underestimating the low flow period while overestimating streamflow at the end of the ablation period in late summer (see Fig. 8g and h). Nevertheless, WaSiM reproduces the seasonal streamflow pattern in the upper Kauner Valley well. It overestimates the mean daily standard deviations (i.e., observed 1.4 m³ s⁻¹ and simulated 1.9 m³

s⁻¹) that are particularly visible in spring and summer (see Fig. 8g). Over the entire simulation period (1973-2015), the daily KGE is 0.76, percent bias -5.5, and RMSE 2.2 m³ s⁻¹, which is satisfying considering that the model is not calibrated, meaning the model parameters are not optimized according to the extended simulation period. Over the shorter period at the end of the simulation run (2007-2015), the daily KGE is 0.72, percent bias -1.1, and RMSE 2.4 m³ s⁻¹. Although daily KEG is slightly better than the ones of the shorter model run, the negative percent bias increases from -0.7 to -1.1 %. Overall, the goodness of

fit criteria of the longer simulation run indicate that the model performance is reliable for a highly glaciated catchment, such as the upper Kauner Valley, over a longer period (> 30 years).

The evolution of annual mean temperature and the consequences for snow and glacier development are presented in Figure 9. The first panel of Figure 9a shows the corrected mean annual WRF temperature evolution of the research area. The vertical red line represents a significant change point in the annual temperature time series detected by the Pettitt test in 1987. This

particular change point has already been observed in previous studies in different parts of the European Alps (e.g., Marty, 2008; Marcolini et al., 2019; Hofmeister et al., 2023).

The simulated cumulated glacier mass balances (Fig. 9b) exhibit a continuous negative trend from the end of the 1980s. Before that, the glaciers were in an equilibrium state or had even slight positive mass balances. The positive glacier mass balances align with observed advances of Gepatsch- and Weissseeferner (Altmann et al., 2020). As the graph shows hourly

simulation results, the seasonal variability of snow accumulation and ice ablation is clearly visible in periods with negative glacier mass balance. The Pettitt test found a significant change point in the summer of 1994. With a delay of six years, a significant change point was detected in the fractional glacier cover (fglc) in 2000 (see Fig. 9c). In general, the evolution of the simulated fglc shows a similar pattern to the cumulated glacier mass balances. The black triangles illustrate years with





**Figure 8.** Mean day of the year of observed (black line) and simulated (blue line) streamflow (left panels) including the standard deviation and empirical cumulated distribution functions (right panels) over four different periods, i.e., 1973-1987 (a, b), 1987-2001 (c, d), 2001-2015 (e, f), and as overall mean (g, h).

observation data. The simulated fglc follows the negative trend of the observations but has a positive offset at the end of the simulation period (i.e., observed fglc is 0.3 and simulated is 0.35).

Figure 9d represents the long-term evolution of simulated snow cover duration (scd) in the upper Kauner Valley. Interestingly, the Pettitt test found a significant change point already in 1985. Before this particular year, the mean scd was fluctuating between 290 and 330 days. After 1985, scd never exceeded 310 days and was even below 260 days in 2002. The significant





reduction of scd is a prerequisite for the negative glacier mass balances, as more firn and ice are exposed to the short-wave
radiation for a longer period, leading to higher melt rates.

**Figure 9.** Simulated mean annual temperature (a), simulated cumulated glacier mass balances as catchment average (b), observed (black triangles) and simulated (blue points) fractional glacier cover (fglc), and simulated annual snow cover duration (d). Significant change points are indicated with the vertical red line.



## 4 Discussion

### 4.1 Single and multivariate correction

It should be mentioned that we are mainly evaluating and correcting the WRF temperature. However, from a physical perspective, a multivariate approach would be better, as the surface temperature controls the relative humidity. We used the 2 m relative humidity from the WRF model in the hydrological simulations, knowing that it is also dependent on the surface properties. The importance of relative humidity is particularly high when physics-based approaches are used for computing evapotranspiration (Penman-Monteith) and snow melt (energy balance). We have not tested the sensitivity of the simulation results concerning the relative humidity from different eta levels. However, this is also beyond the scope of this study, as relative humidity was only simulated and stored at 2 m altitude by the WRF model, and rerunning the WRF model with the given setup and domain requires heavy computational resources for several months. Another important meteorological variable is shortwave incoming radiation that was dynamically corrected by the WaSiM model concerning topography and day of the year. WRF wind speed originates in our study from 10 m above the ground and was not adjusted to the finer topography of the WaSiM model. Since a dynamical downscaling of wind speed and direction requires a complex model approach (e.g., Wagenbrenner et al., 2016; Quéno et al., 2023), we have refrained from doing so, also knowing that the wind speed has no influence on the snow redistribution in WaSiM. Wind speed mainly influences the simulation of snow melt and evapotranspiration rates in WaSiM. Since WaSiM only considers one dominant wind direction for the wind-induced snow redistribution, a time-dependent forcing with actual wind directions is not yet possible.

### 4.2 Quality of WRF precipitation

Although the main focus of this study is on the spatiotemporal WRF temperature distribution for high-resolution distributed hydrological model simulations in alpine catchments, the quality of WRF precipitation is essential for reliable and plausible HM results. A key aspect is the removal of the "drizzle bias", meaning the WRF's tendency to overestimate the occurrence of light precipitation events. This bias was already observed and discussed in other studies (e.g., Lazoglou et al., 2024; Velasquez et al., 2020). These low precipitation events have only a low impact on the annual water balance. In our study, the removal of low precipitation events results in a reduction of annual precipitation of only 4 %. Since WaSiM considers the presence of precipitation in the estimate of cloudiness, the precipitation frequency can indirectly impact other processes, such as the computation of the long-wave incoming radiation (Schulla, 2021). We remove the low precipitation events by monthly thresholds (Tab. 2) under the assumption that these are stationary over the period 1971 to 2015. However, this assumption is only partly true as the observed monthly precipitation days at Vergoetschen station show an increasing trend (Fig. S8) for the first ten years (i.e., 1971 to 1981). After 1981, the sum of monthly precipitation days seems to be stationary at this site until 2015. Nevertheless, the interannual variability of monthly precipitation days is well visible (Fig. S8). In principle, we could remove the low precipitation events for each year separately, as the precipitation observations from Vergoetschen station overlap with the period of the longer simulation run (1973 to 2015). However, as we are planning to perform an even longer simulation starting in 1850, we have deliberately opted for global monthly thresholds. From our perspective, seasonal or even





sub-seasonal threshold values are needed for the removal of low precipitation frequencies due to the clear tendency concerning
observed monthly precipitation days between summer and winter (Fig. S7).

Apart from the precipitation frequency, we have not further investigated the quality of the WRF precipitation, which is a key
driver and source of uncertainty in the HM. The evaluation of the precipitation timing, magnitudes, and intensities is partic-
ularly limited by station observations due to the high spatial heterogeneity and, most importantly, by the large measurement
uncertainty in high-elevation regions (> 2000 m a.s.l.). The undercatch of precipitation measurements in cold regions was
demonstrated in several studies (e.g., Kochendorfer et al., 2016; Mair et al., 2016). A potential approach for evaluating the
WRF precipitation magnitudes would be in an indirect way by closing the water balance, presupposing that observed stream-
flow is available. However, this is only possible when there is no significant contribution from glacier melt and permafrost
thawing, which is usually unknown. Due to the many sources of uncertainty, we recommend, in particular, evaluating the
seasonal precipitation of RCMs in glaciated high mountain regions with seasonal observation data from glacier mass balance
programs, as an evaluation only against streamflow is problematic due to the delayed response and the lack of information on
the remaining water balance components such as evapotranspiration and sublimation. There are only a few glacier observa-
tories in the European Alps with long temporal data availability, including streamflow observations. The Glacier Observatory
at Vernagtferner would be a good study site for evaluating seasonal precipitation data sets (WGMS, 2025). Unfortunately,
interdisciplinary and long-term observations of cryospheric and hydrological processes are still very limited in high-elevation
catchments. Nevertheless, the correction of rainfall intensities remains problematic under the aspect of limited observation data
and non-stationary conditions (Teutschbein and Seibert, 2012).

### 4.3 Representation and evolution of land use categories in WRF

The question arises from the application of more and more high-resolution RCMs in mountainous regions, whether the repre-
sentation of land cover properties and topography is sufficiently resolved for detailed snow-hydrological studies. Tomasi et al.
(2017) observed that WRF reproduces rather poorly near-surface temperature (2 m) over snow-covered terrain in the Adige
Valley (eastern Italian Alps), with an evident underestimation, during both daytime and nighttime. The main cause of these
errors lies in the miscalculation of the mean grid cell albedo, resulting in an inaccurate estimate of the reflected solar radiation
calculated by the LSM Noah. Therefore, Tomasi et al. (2017) modified the initialization, the land-use classification, and LSM
are performed to improve model results, by intervening in the calculation of the albedo, of the snow cover, and of the surface
temperature. By doing so, a significant improvement in the comparability between model results and observations is achieved.
This means outgoing shortwave radiation is lowered, 2 m temperature maxima increased accordingly, and ground-based ther-
mal inversions are better captured (Tomasi et al., 2017).

Besides the spatial representation of these characteristics, another question is whether the long-term evolution of the surface
properties is considered or not. Due to the increasing greening effect (i.e., increase in vegetation productivity) and the fast retreat
of glacier cover in the proglacial areas (Rumpf et al., 2022), constant land cover classes cannot account for the accelerated
change of the surface-albedo properties. Especially for long-term (> 30 years) climate-related studies, a constant land cover
parameterization may lead to implausible behavior of some important meteorological variables (e.g., temperature) that are





particularly sensitive to the surface albedo. This behavior will impact all following analyses, as demonstrated in this study. Hence, there is a need for either a robust workaround, as we introduce in this study, or the representation of a dynamic land
cover. The latter is particularly problematic as land cover information, for example, of the Alpine region, is scarce before 1950 due to the lack of remote sensing data, and the reconstruction of earlier stages is particularly complex (Ramskogler, 2023). Hence, a robust workaround might be the most suitable approach for most longer RCM applications in snow and glacier-dominated areas.

## 4.4 Uncertainties in the model cascade

Uncertainties are occurring and affecting all parts of the complex model cascade that was applied in this study (Jobst et al., 2018). The widely used global atmospheric reanalysis product (20CRv3) only assimilates surface pressure data that ensures temporal homogeneity over the entire reanalysis period (i.e., 1836–present), but the limited input data strongly influences variables such as precipitation or near-surface temperature, as they are not directly constrained by observations (Slivinski et al., 2019). Another source of uncertainty is the parametrization and handling of the boundary conditions of the WRF model. To
reduce the uncertainty in the dynamical downscaling from the global reanalysis product (20CRv3) to WRF, a nested approach was chosen that considers three different spatial domains and resolutions (18, 6, 2 km). Although the 2 km WRF can resolve convection and orographical effects in the central European Alps, the resolution is still too coarse for a plausible representation of temperature and precipitation patterns in high-elevation catchments. As demonstrated in this study, considerable implausible behavior originates from the direct use of the 2 km WRF data as model forcing for a high-resolution (i.e., 25 m) HM. Based
on this observation, we, therefore, conclude that space-related uncertainty (including elevation gradients) between WRF and WaSiM is the dominant uncertainty source in the applied model cascade. One plausible argument is that a statistical bias correction would remove the cold bias in the 2 m WRF temperature. However, this requires long-term and continuous temperature observations at higher altitudes (> 2700 m) that are particularly scarce and represent only in situ information. In addition, meteorological observations are highly affected by errors and failures at these altitudes (Hofmeister et al., 2023). In several studies
(e.g., Rangwala and Miller, 2012; Kuhn and Olefs, 2020), the higher elevated regions have experienced a stronger warming, which makes the statistical bias correction only relying on lower elevated stations questionable.

We have refrained from a comprehensive calibration of the HM parameters as our intention was to demonstrate the impact of RCM forcing data on the model results. Hence, we used a standard parameterization that relied mainly on values from the literature. Usually, parameter optimization algorithms try to compensate for errors in the model forcing by selecting extreme
and implausible values of the parameter ranges (e.g., for the rain-snow threshold), especially when applied in a single-objective calibration. A multi-objective optimization approach will most likely fail to provide reasonable and plausible model results for the individual processes, such as snow and glacier dynamics and the interaction with the streamflow generation (Tiel et al., 2022; Schaffhauser et al., 2024a) under the assumption of a biased model forcing.

Another major uncertainty in the HM setup is in the initialization of the glacier volume, especially in highly glaciated
catchments such as the upper Kauner Valley. Due to a lack of glacier data in 1968, we had to rely on published values to parameterize the VA scaling factor (Chen and Ohmura, 1990). Another uncertainty originates from the conceptual glacier





routine implemented in WaSiM, which does not consider actual ice flow dynamics for balancing masses between accumulation and ablation areas. This is particularly problematic if the model is applied for longer simulation periods, as demonstrated by Pesci et al. (2023). We observed a delayed recession in the simulated fractional glacier cover (Fig. 9c), which is most likely an
effect of the simple glacier representation in WaSiM. In general, it is well known that simplified glacier routines have issues when they are applied for periods with positive glacier mass balances and glacier advance (Seibert et al., 2018; Schaffhauser et al., 2024b).

The date and length of the initialization period slightly impact the simulated annual streamflow. When starting WaSiM in 1969, the annual streamflow was 1.1 % lower at the end of the simulation period compared to the simulation that started in
2006. Together with the consistent goodness of fit criteria, this indicates that the WRF WaSiM setup provides reliable results even after 42 years of simulation period. Consequently, these results show that the HM provides essential information on the performance and reliability of downscaled meteorological reanalysis data for high mountain areas. Process-based evaluation of the HM results provides comprehensive information on the quality of bias correction methods and should therefore be an integral part of bias correction frameworks for RCMs.

A general disadvantage of the presented model cascade from RCM to HM is the high computational effort that comes with the dynamical downscaling with the WRF model. Although the research catchments of interest have a relatively small size of about 60 km$^2$, a significantly larger area needs to be modeled with the WRF model. Therefore, simplified three-dimensional atmospheric models, such as Berg et al. (2024), are more efficient for the dynamical downscaling of RCM data to the catchment scale. The coupling of these simplified atmospheric models with distributed snow-HMs would allow for sensitivity runs and
projection runs with several ensembles due to the reduced computational time. This coupling is, therefore, a promising direction for further development in the field of climate-related studies of the cryo- and hydrosphere interactions.

### 4.5 Integrated hydrometeorological modeling systems

A potential solution for the temperature and precipitation scale change issue could be an integrated hydrometeorological modeling system, like WRF-Hydro. WRF-Hydro can be fully coupled with the WRF atmospheric model, allowing for dynamic
representation of precipitation, temperature, and energy inputs. WRF-Hydro allows for variable spatial resolution in the hydrologic model, enabling refinement in river routing while maintaining coarser land surface or atmospheric resolution (Gochis et al., 2025). When using Noah-MP LSM, WRF-Hydro can include multi-layer snow physics, allowing for improved snow accumulation and melt simulations (Eidhammer et al., 2021; Gochis et al., 2025). Besides the high potential, there are certain limitations of the WRF-Hydro model, particularly in its default configuration. For example, WRF-Hydro does not include
glacier dynamics, mass balance, or ice flow processes. Processes like snow sublimation, wind-driven redistribution, and gravitational slides are not well captured, leading to inaccuracies in snowpack modeling (Eidhammer et al., 2021), especially in the complex terrain of high-elevation catchments. The spatial resolution of the hydrologic model grid is typically in the range from 100 m to 1 km (Eidhammer et al., 2021; Gochis et al., 2025), which may misrepresent subgrid river networks in steep terrain, as the smoothing of the DEM can remove important terrain features.



Eidhammer et al. (2021) incorporated a one-dimensional column snowpack model (Crocus) into the WRF-Hydro to allow for direct surface mass balance simulation of glaciers and subsequent modeling of meltwater discharge from glaciers. The integration of the Crocus snowpack model led to a more realistic simulation of surface albedo over glacierized grid cells, as surface albedo is represented by snow, where there is accumulated snow, and glacier ice, when all accumulated snow is melted. Consequently, the WRF-Hydro/Glacier system reproduced observed glacier mass balances and glacier runoff well. Neverthe-

less, the WRF-Hydro/Glacier system still does not address glacier dynamics nor wind redistribution of snow (Eidhammer et al., 2021).

     Another critical fact for the application of the WRF-Hydro on small spatial scales (< 100 m) is that the WRF atmospheric model does not dynamically downscale the surface energy balance and precipitation on the finer grid resolution of the hydrologic model. This means an additional bias correction or downscaling is needed when the hydrological model is applied on a

fine grid resolution in a complex terrain (Eidhammer et al., 2021). Hence, the approach proposed in our work represents a step forward and an alternative for modeling long-term hydrological changes on a small scale (< 100 m) in glaciated high-elevation catchments.

## 5    Conclusions

The objective of the presented study was the development and evaluation of a workflow for transferring coarse-resolved 2

km RCM data to a much finer spatial resolution (i.e., 25 m) of a fully distributed hydrological model, by which long-term climate-related hydrological changes in a highly glaciated catchment in the European Alps were investigated. As RCM, we used the WRF model for dynamically downscaling a global reanalysis product with three nested domains over the central European Alps. When comparing the WRF 2 m temperature with in situ stations from the study site, the upper Kauner Valley, we noticed a negative bias for the highest station (> 2600 m a.s.l.). By further analysis, we found that the negative bias is due to

the relatively coarse representation of surface properties controlling the land-atmospheric interactions, in this particular case of snow and glacier cover in the WRF model. Therefore, we developed a workflow for extracting hourly temperature lapse rates from the WRF atmosphere in the range from 17 m to 6000 m above the surface, as this part of the WRF atmosphere is relatively insensitive to the surface properties. Due to a lack of highly elevated meteorological stations (> 2700 m a.s.l.), we evaluated the plausibility and reliability of the corrected WRF surface temperatures in a two-stage process. First, against a set of four in

situ stations that were covering an altitudinal gradient of almost 900 m. Second, by forcing a physics-based hydrological model (i.e., WaSiM) over two periods with six and 42 years of simulation time with an hourly time step and critically evaluating the model performance against several observation products (e.g., streamflow, in situ SWE, fractional snow covered area, and glacier cover).

     The comparison of hourly WRF corrected temperature frequencies with in situ observations proved a high agreement and

was better represented than by the uncorrected WRF temperatures. When forcing the WaSiM model with the uncorrected WRF temperature, WaSiM failed to reliably simulate cryospheric and hydrological processes (i.e., daily KGE 0.42 and percent bias -46) in the Kauner Valley. The corrected WRF temperature resulted in much better model performance concerning seasonal





dynamics of streamflow (i.e., daily KGE 0.71 and percent bias -0.7), fractional snow covered area, and glacier mass balance. Even over a longer simulation period of 42 years, the WaSiM model provided good streamflow predictions with a daily KGE
of 0.72 and a percent bias of -1.1 at the end of the simulation period (2007-2015).

We tested the transferability and consistency of the long-term temperature trend of the corrected WRF temperature by comparing monthly and annual temperatures for the period 1973 to 2015 with homogenized station data of the HISTALP data set. The Obergurgl station is located in a neighboring valley (i.e., the Ötz Valley) approximately 20 km away from the Kauner Valley at 1936 m a.s.l.. Both temperature time series had a significant positive trend (i.e., station 0.06 °C and WRF corrected
0.05 °C) over this particular period, with the corrected WRF product slightly underestimating the annual warming by 0.01 °C. Interestingly, we found a stronger warming of 17 % at the highest location of the Kauner Valley (i.e., 3800 m a.s.l.) compared to the lowest station site (i.e., 1770 m a.s.l.).

Our study provides valuable insights into the complex process interdependency of the cryo- and hydrosphere of highly-glaciated catchments and the need for an accurate spatiotemporal temperature representation in such a rugged topography.
The still relatively coarse-resolved RCMs cannot resolve the small-scale heterogeneity of the topography and land surface properties. Hence, the developed workflow of this study provides an alternative to classical temperature bias correction methods (e.g., quantile mapping) that mainly depend on the availability and quality (temporal homogeneity) of observation data. In addition, the proposed method of deriving TLRs from the free WRF atmosphere is physically consistent, transferable to other sites, independent of non-stationary conditions, and can be applied for future climate impact studies.

*Code and data availability.* A compiled version of the hydrological model *WaSiM* is available at http://www.wasim.ch/en/ for Windows and Linux OS (no source code available). The ΔSnow model (v1.0.2) used for transforming the snow height data to snow water equivalent is accessible in the R package "*nixmass*" (Winkler et al., 2021). The modified Mann-Kendall test after Hamed and Rao (1998) is available in the "*modifiedmk*" R package (version 1.6 https://cran.r-project.org/web/packages/modifiedmk/modifiedmk.pdf). The Kolmogorov–Smirnov (KS) test is available in the base R code (version 1.5.1 https://www.rdocumentation.org/packages/dgof/versions/1.5.1/topics/ks.test). The
Pettitt test is available in the "*trend*" R package (version 1.1.6 https://www.rdocumentation.org/packages/trend/versions/1.1.6/topics/pettitt. test). The glacier outlines are available from the Austrian Glacier Inventories for 1969 (GI 1), 1998 (GI 2), 2006 (GI 3), and 2015 (GI 4) on PANGAEA. The digital elevation model of the entire Tyrol can be downloaded in 5 m or 10 m spatial resolution (https://www.data.gv. at/katalog/dataset/land-tirol_tirolgelnde#additional-info). The homogenized temperature data at the Obergurgl station are obtained from the HISTALP data set (https://www.zamg.ac.at/histalp/). The precipitation records of the Vergoetschen station were downloaded through the
Austrian data portal eHYD (https://ehyd.gv.at/). The temperature records of the Vernagt Pegel station are available on PANGAEA (https://doi.pangaea.de/10.1594/PANGAEA.829530). The other hydrometeorological data of the upper Kauner Valley are only available from TIWAG on request. The WRF data of the Kauner Valley domain for the period 1969 to 2015 will be uploaded to an online repository as soon as the manuscript is accepted for publication.



*Author contributions.* All authors participated in the conceptualization, methodology, and design of the study. MP performed the WRF sim-
ulations and carried out the first analysis of the WRF data. FH carried out the formal analyses and investigations, including the hydrological
modeling. FH wrote the original manuscript and visualized all the results, and all co-authors contributed to the editing and review. BS and
GC supervised the work and acquired the project funding.

*Acknowledgements.* The authors acknowledge the valuable discussions with Marco Möller about the methodology and results of this study.
The authors thank the project partners of the Eurac research for providing the information on the catchment land cover. The authors thank
the TIWAG-Tiroler Wasserkraft AG, HD-Tyrol, and GeoSphere Austria for the hydrometeorological observation data. FH, XF, and GC
acknowledge the support of the German Research Foundation (DFG) research unit (FOR2793/2) investigating the "Sensitivity of High
Alpine Geosystems to Climate Change since 1850" (SEHAG) under grant CH981/3-2. MP and BM are also part of the SEHAG DFG
research unit and acknowledge the grant numbers MA 6966/4-2 and LA4426/1-2. XF and BS acknowledge the Swiss National Science
Foundation (SNSF) grant (200020E-204030) for "SEHAG Subproject 2 -Impact of climate change on groundwater storage in high Alpine
catchments: from observation to model predictions".

*Competing interests.* The authors declare that there is no competing interest in this work.



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
