# Peer review of "The impact of small-scale surface representation in WRF on hydrological modeling in a glaciated catchment"

_EGUsphere, 2025_

## Author Comment (AC4)

**Final response to RC1**

**General comments**

Hofmeister et al. applied WRF 2 km data to a physically based hydrological model with a resolution of 25 m resolution in a small (62 km2) highly glaciated catchment in Austria. They provided two different types of air temperature to the hydrological model, one is the temperature 2 m above the surface, and one the so-called "corrected" temperature, which was derived from WRF temperature on the first and 20th level (~3 km above ground). They also corrected a drizzle effect in WRF by setting all values to 0 below a monthly determined threshold based on a station nearby the study area.

Comparison with four stations within the catchment showed an improvement for the corrected temperature, especially for the highest station which was presumably wrongly classified as glaciated in WRF. With the corrected temperature also the hydrological model performance improved, e.g. comparison with MODIS determined snow covered fraction improved, the sign of glacier mass balance turned from obviously wrong positive to realistic negative and comparison with observations of streamflow improved.

However, the authors promised in the title, in the abstract and the introduction with the stated research questions much more than this synthesis, which was later not presented in the manuscript. I will provide examples in the next section. Oppositely, the manuscript contains significant parts which do not relate to the stated research questions and rather show interesting applications of such a model chain or a discussion/review on other studies. It is a long manuscript and there is a clear need to focus.

While reading the manuscript it was often not clear to me what motivation was or how the methods were applied. Reading the conclusions at the very end was a (belated) eye-opener for me, and yet questions remain (see next section for details).

Another important issue is the limited availability of data to ensure that the hydrological model used to evaluate the altered temperature inputs does not simply improve the results due to error compensation (see details in also in next section).

I acknowledge that preparing WRF model results and applying a high-resolution, physically based hydrological model as WaSIM is an extensive task (WRF results were probably taken from Altmann et al. (2024) and it is not clear if they were independently generated, but this is not the relevant point here). I suggest that the authors completely revise their manuscript and clearly focus on what they can provide with the presented analysis. However, I have the feeling that a distilled manuscript is not sufficient for an own publication and that the authors should rather proceed with their obvious plans of analyzing model output from long-term runs since 1850, where this here presented part of adjusting air temperatures is a small paragraph.

We are grateful for the comprehensive feedback on our manuscript. We agree that the manuscript is relatively long and extensive. This originates from the fact that the application of such a model cascade from an RCM to a distributed hydrological model is rather complex when cryo-hydrological processes are modeled with physics-based model approaches. Consequently, the method and discussion sections of

the manuscript are relatively detailed, providing descriptions of the correction of WRF temperature and precipitation data, as well as their respective limitations, which are discussed in the discussion section. We apologize for the confusion, as the manuscript was intended to be submitted as a technical note rather than a research article. Nevertheless, we agree that a revised manuscript should clearly link the title, abstract, introduction, and synthesis to enhance comprehensibility. Additionally, we will condense the manuscript to its core aspects, remove unnecessary content, or shift it to the supplement when needed as additional information.

Although the intention of the developed workflow is to model long-term hydrological changes from 1850, we focus on a more recent period (1970-2015) in the presented manuscript to evaluate the model results against cryo-hydrological observations. As the manuscript is already extensive, we have refrained from including the long-simulation results that start in 1850. To avoid any confusion regarding the periods of interest, we will focus solely on the evaluation period (1970-2015) in the revised manuscript.

As the developed methodology for correcting and evaluating the WRF temperature is rather complex and, from our perspective, novel in the field of meteohydrological model cascades, it is not possible to briefly describe the applied workflow in a single subsection. However, it would be conceivable to integrate a large part of the technical descriptions and evaluations relating to the WRF corrections into the Supplement.

**Detailed comments**

Stated but not done (as it first appears)

Small-scale surface representations in WRF

The title and the first research question (line 96) suggest that small-scale surface representations in WRF and their impact on subsequent hydrological models have been studied. The authors want to present the effects of an alternative derivation of near surface air temperature, which should consider elevation and land cover classes (line 100). However, this correction is not based on any small-scale surface properties; the opposite is true: It is removing the influence of the probably too coarse WRF land cover information by simply averaging temperatures over the whole domain at elevation levels where they are relatively insensitive to land surface properties (which is at the end also stated in the conclusion, lines 684-687).

We agree on the misleading terminology and the conflict between the title/abstract and the conclusion. Our motivation is to first demonstrate the impact of WRF land cover classes on WRF near-surface (i.e., 2 m) temperature and secondly to provide a workflow for adjusting the WRF temperature, which is independent of the land cover. Since the spatial WRF land cover resolution cannot resolve the small-scale surface characteristics of a meteorological station, which is typically a few square meters in size, a robust and consistent workflow for adjusting WRF temperature is highly needed, especially in the complex terrain of high-elevation catchments. Our study demonstrates that spatial averaging of WRF temperature from different eta (i.e., pressure) levels still yields more reliable temperature predictions at several meteorological sites located at different elevations compared to the initial 2 m WRF temperature.

**Bridging the scale from WRF to WaSIM**

In the abstract the authors stated that the main challenge is to transfer forcing data to the much finer resolution of the hydrological model. The authors wanted to present a workflow for bridging this scale. This relates to the second research question in the introduction (lines 97-98). However, I can only see the application of rather simple WaSIM integrated algorithms (lines 300 to 309) which cannot be meant by a newly presented workflow.

Unfortunately, the wording in the abstract is too vague, causing confusion. It is true that the applied algorithms for inter- and extrapolation of meteorological variables are already implemented in WaSiM and were not developed in this study. We will revise the sections accordingly.

**Consistency of bias-corrected RCM data over longer periods**

This relates to the last and third research question (lines 98-100). However, the authors have not applied bias-correcting method for temperature using observations. The temperature adjustment presented is not using observations (but still removed biases). The consistency was tested in comparison to a nearby valley. However, this was not done in relation to non-corrected data (and probably only for one reference grid point), so one cannot state an improvement.

Yes, that is true. We will also include the uncorrected WRF temperature in this comparison for consistency. Yes, it only represents one meteorological station and the respective grid cell. However, we can perform further evaluations for other stations with long temperature time series from other valleys and add them to the supplement.

For precipitation they removed a drizzling bias using long-term observations which is a classic biascorrection case. However, no consistency results were presented here.

We only present evaluations of the monthly precipitation days in the Supplement (Figure S8). We will extend the evaluation and add some consistency results to the Supplement.

**Modelling period**

In the abstract it was stated that the modelling period is from 1850 to 2015, however, only data from the 1973 to 2015 are analysed in this manuscript.

As previously mentioned, we will focus solely on the period from 1973 to 2015 in the revised manuscript.

**Topics outside of the focus**

The manuscript can substantially be reduced to topics which are part of the research questions. Example of topics, which are not part of an evaluation of presented temperature and precipitation correction may be the determination of significant change points (unless it is not evaluating the corrected input).

Another example is the long discussion as it does not relate own results with other studies. It can be shortened and put in the introduction or placed in a review paper. The implications of own results are hardly discussed in the discussion.

We will condense the manuscript to its core aspects, remove unnecessary content, or shift it to the supplement when needed as additional information. We will set the main focus of the discussion on our own results.

**Clarity**

In equation 5 and 6 it is stated how the corrected and uncorrected temperature at a certain station location is calculated. However, it is unclear to me how this is done for WaSIM grid points: In lines 302-304 I can read some information, but how the variable lapse rate is determined is unclear, and whether this is done for both variants. This is particularly relevant as Eq. 6 contains a fixed lapse rate to account for elevation differences.

WaSiM computes a time-step-dependent (i.e., hourly in our study) elevation-temperature gradient based on the constant elevation of the WRF grid cells and the respective WRF temperature and assigns each WaSiM grid cell a temperature according to its respective elevation. For consistency, this approach is identical for uncorrected and corrected WRF temperature. We will add another sentence after line 304 to clarify this. The fixed lapse rate from Eq. 6 is only used for comparing corrected and uncorrected WRF temperatures at station locations, but it is not considered in the WaSiM simulations. As already mentioned, the temperature lapse rate is not fixed for both WaSiM simulations (i.e., corrected and uncorrected).

Equation 5 and 6 differ in two ways: First the mentioned lapse rate is fixed and second the 2 m temperature is used in Eq. 6 for the uncorrected version, while Eq. 5 used an hourly changing lapse rate and the first level temperature. It is unclear to me, which part is the relevant one for improvements. As this is the core part of the paper, I would suggest some additional analysis.

As we are primarily evaluating and comparing the ECDFs of uncorrected and corrected WRF temperatures with station observations (see Fig. 4), the expected difference should not be significant, regardless of whether it involves a fixed lapse rate. Nevertheless, we will add some additional evaluations regarding the sensitivity of the fixed lapse rate on the uncorrected WRF temperature at station locations. From our understanding, the main difference between WRF temperature computed with Eqs. 5 and 6 is due to the different interaction with the WRF land use classes (see Fig. 4). As mentioned before, the fixed temperature lapse rate is not considered in the WaSiM simulations.

One main argument for using the corrected temperature approach was that the land cover classification too coarse in WRF (conclusions lines 684 - 686). The authors missed to show a map of this classification.

Yes, we will add a map showing the WRF and WaSiM land cover classes to the Supplement.

**Improvements or error compensation?**

In the conclusion the authors state that due to the lack of meteorological stations above 2700 m they evaluate the impacts of the corrected temperatures with a hydrological model. However, it is unclear to me if the authors have included not more issues with this attempt than less. The hydrological model certainly includes uncertainties in the modelling cascade, which are also discussed in the manuscript.

One issue is that the study area is not really suited for this evaluation. While I see the point that there is a switch from an unrealistic to a realistic sign in the glacier mass balance trend, there are no measurements of glacier mass balances available. There is also no spatial snow distribution data used in this study. So, the reader does not know whether the shown improvements in fractional snow-covered area, glacier mass balance and runoff comes not from error compensations elsewhere in the hydrological model.

Before forcing the hydrological model WaSiM with WRF data, we first ran the model using meteorological station data. We evaluated its performance using a process-based approach (similar to the one employed in this study). The calibration and validation period was from 2006 to 2021. This ensures that WaSiM delivers reliable results for the Kauner Valley. The results are not included in this manuscript and supplement, as they would increase the complexity of this study. We used the same parameterization and only changed the meteorological input data (i.e., regional climate data from WRF) to examine its plausibility. We should ensure that the hydrological model gives reasonable results with these new meteorological inputs. Previous studies have demonstrated (e.g., Hofmeister et al., 2023; Clerc-Schwarzenbach et al., 2024; Clerc-Schwarzenbach et al., 2025) that meteorological forcing data can be one of the primary source of uncertainty, leading to implausible hydrological model results. The evaluation with a hydrological model is an essential step to validate meteorological / climate products for use with a hydrological model, but is often omitted. The result of this evaluation says whether the product is compatible with the hydrological model, but it does not necessarily assess the "absolute" value of that product for other applications.

Our assumption is that high-quality forcing data should enhance the reliability of hydrological model results even if the hydrological model is uncalibrated. We refrained from a forcing-specific calibration of the hydrological model due to the risk of compensating for the actual differences between the meteorological / climate products. In addition, we performed a process-based evaluation (e.g., streamflow, SWE, snow cover) that enables the reduction of potential compensation in the model, allowing for a more comprehensive approach beyond focusing solely on streamflow.

Although the Kauner Valley lacks spatial snow and glacier observations, it remains a unique research site with continuous streamflow and meteorological observations dating back to the 1970s. Since the bias problem arises particularly for high-elevation catchments with complex topography, unfortunately, these catchments still contain glaciers.

The authors argue with Figure 7 that the improvements stem from air temperature corrections leading to an improved distribution of snow, to an improved snow and glacier melt and finally to an improved runoff. One indication for an error compensation is the largely biased snow water equivalent (SWE) as well as its insensitiveness to the temperature correction (Fig 7a) at one single station. I think the authors need to provide additional insights in the correctness of their argumentation. For this it may be relevant to evaluate the model chain in regard to a realistic SWE distribution. To my opinion a 500 m resolution MODIS product is not sufficient for this complex terrain, which was also stated by the first author in Hofmeister et al. (2022). Such a study can be done using Sentinel2 data as done by Hofmeister et al. (2022) but needs to be presented for WRF input and for this specific catchment but may be limited to a few winter seasons.

We completely agree with the limitations of the MODIS snow cover map for evaluating the performance of the WaSiM snow module. We would have also preferred to use snow cover data based on Sentinel 2, but unfortunately, there is not a sufficiently long overlap between the data, as the WaSiM simulation ends in 2015, the year Sentinel 2 was launched. Therefore, the comparison between MODIS and simulated snow cover is still better than no evaluation at all. We will add this constraint to the discussion.

**References**

Clerc-Schwarzenbach, F., Selleri, G., Neri, M., Toth, E., van Meerveld, I., and Seibert, J.: Large-sample hydrology – a few camels or a whole caravan?, Hydrol. Earth Syst. Sci., 28, 4219–4237, https://doi.org/10.5194/hess-28-4219-2024, 2024.

Clerc-Schwarzenbach, F. and do Nascimento, T. V. M.: Evaluating the quality of the E-OBS meteorological forcing data in EStreams for large-sample hydrology studies in Europe, EGUsphere [preprint], https://doi.org/10.5194/egusphere-2025-3710, 2025.

Hofmeister, F., Graziano, F., Marcolini, G., Willems, W., Disse, M., and Chiogna, G.: Quality assessment of hydrometeorological observational data and their influence on hydrological model results in Alpine catchments, p. 02626667.2023.2172335, https://doi.org/10.1080/02626667.2023.2172335, 2023.

---

## Author Comment (AC5)

**Final response to RC2**

Review of The impact of small-scale surface representation in WRF on hydrological modeling in a glaciated catchment by Hofmeister et al., submitted to HESS.

The manuscript (now being considered as a technical note - communication with editor on 29-09-2015) principally addresses the implications of considering the WRF land-surface classification upon the distributed air temperatures for input to a hydrological model using a glacierised catchment in the Austrian Alps. The impact of re-deriving air temperature lapse rates from pressure levels of a 2km WRF model is assessed at a very high model resolution of 25m using an intermediate complexity hydrological model with consideration of glaciers and snow. The authors highlight that the representation of glacier mass balance, snow cover dynamics and catchment hydrology are all improved with the representation of air temperature which acts more independently of the prescribed land cover class from the 2km WRF grids.

The quality of the writing and figures is generally very good and the arguments are reasonably supported for the most part. The impact of the temperature downscaling methodology for long-term glacier mass and related impacts on the catchment hydrology are interesting and potentially useful for the community. The actual long-term modelling results are very nice (despite not being calibrated or reasonably evaluated) and could be very interesting for the community.

My initial feeling upon reading was, however, that the manuscript does not provide a substantial advance with respect to catchment modelling and the work rather describes a methodological approach to improve the WRF forcing data (that is not always so clear and well justified), albeit in a very long article. With this manuscript now being considered as a technical note, the authors must substantially shorten the work and focus on a clear, yet concise argument for the practical applications of air temperature lapse rates from WRF for catchment modelling, ideally testing different combinations of lapse rates/methods and providing a much clearer description of the method and the context for its relevance. The change of article type already constitutes a major revision for the authors, which is consistent with my recommendation, before being acceptable for publication in the journal. My major and minor technical comments are written below.

Thank you very much for your detailed and helpful feedback on our manuscript. Firstly, we would like to apologize for the confusion regarding the article type. In fact, the submitted manuscript is a technical note and not a research article. A significant amount of time and effort has already been invested in developing and testing the workflow for enhancing hydrological simulations using WRF forcing, which is also reflected in the manuscript's length. We are confident that a revised manuscript incorporating your comments will make a significant contribution to the scientific community.

**Major Comments**

1. The manuscript is long and overly descriptive in places in order to present a methodological step toward improving forcing for a hydrological model. However, a complete analysis of why and where the hydrological model is improved using the lapsed free-air WRF temperatures is still lacking in places. There is no testing of alternate lapse rate approaches or using different pressure level combinations from the WRF, nor whether 'ignoring' the WRF near-surface temperature attribution due to landcover type is in any way dependent upon certain conditions or times of year. The transition to a technical note for the journal should see shortening of other sections describing the WRF and WaSIM modelling, where appropriate, especially where details can be provided in summary tables (main text or SI).

We will reduce the content while strengthening the evaluation part of the manuscript. Additionally, we will further elaborate on why and where the hydrological model is improved using the lapse rate from the free-air WRF temperatures.

Our assumption is that the sensitivity of temperature lapse rates depends significantly on the first eta level (~17 m above ground), as there is a much more homogeneous transition at the subsequent levels, as shown in Figure 1 (Fig. 3 in the manuscript). We can calculate the different lapse rates based on different combinations of eta levels and add them to the Supplement.

Figure 1. Mean WRF temperature lapse rates in  $^{\circ}$ C /100 m for two pairs of consecutive eta levels and averaged over the Kauner Valley. The red line represents the mean lapse rate between eta levels 1 and 20.

2. Regarding the methodological details for the derivation of the lapse rates from pressure levels of WRF, the details are actually lacking still, and it's not clear to me exactly how or why the lapse rates have been derived the way they have. For example, from my reading, the authors average temperatures (and geopotential heights) for each level for each time step (equation 1+2), but then use only a linear fit for the 20th and 1st levels (equation 3), and then make an average of all hours (equation 4). The authors

make the claim that the spatial averaging on each level makes the lapse rates more robust, though with no real evidence of this. This may be negligible due to the size of the catchment, but the reader has no way to understand if the land surface variations (in this case, most of the glacier cells are to the south of the catchment) could still have some impact on the lapse rates and comparison to the observations and impact on the hydrological model etc. The information in figure 3 also highlights that lapse rates are calculated for each elevation band, but it is unclear how that was derived (stepwise lapse rates?), whether taking different eta levels or removing the lowest level (which retains the impact of surface exchanges of WRF) impacts the results of the study here, and potentially for other applications. The authors need to make a clearer case to justify what lapse rates they apply, how robust their approach actually is, and how their approach might succeed or struggle in other, potentially larger, domains. The true value for the community is therefore minimal at the current time.

We will strive to provide a clearer explanation of how we performed the calculation and the reasoning behind it. We first perform spatial averaging for temperature (Eq. 1) and geopotential height (Eq. 2) at each eta level. The actual lapse rates between eta level 1 and 20 are computed with Eq. 3. The temporal averaging in Eq. 4 is only considered in the evaluation of mean lapse rates over longer periods (e.g., for 1970-2015). These temporal averages are not considered in the hydrological modeling, as the model assimilates the corrected and uncorrected hourly temperatures. The uncorrected WRF temperature provides the elevation of the WRF grid cells to the hydrological model, whereas the corrected WRF temperature already represents the actual station elevations computed with Eq. 5.

We generated a flowchart to illustrate the processing of the corrected WRF temperature from the different Eta levels. We hope that the flowchart improves the comprehensibility of this section. Since the current order of Eq. 4 and Eq. 5 is a bit confusing, as it suggests that Eq. 4 is used to calculate the hourly lapse rates, we will swap these two equations in the manuscript.

Figure 2. Flowchart to illustrate the various steps for computing the corrected WRF temperature from eta level 1 and 20.

We agree on the missing proof that the spatial averaging of temperature (Eq. 1) and geopotential height (Eq. 2) for each eta level makes the TLRs more robust regarding spatial heterogeneity. The eta levels are less dependent on the WRF land cover types when computing the spatial average, which depends on the catchment domain. We will rephrase this sentence for clarity.

The calculation of the lapse rates shown in Figure 1 (Fig. 3 in the manuscript) is based on two pairs of consecutive eta levels (e.g., 1 and 2). Accordingly, it is the mean lapse rate of two consecutive eta levels and of all grid cells in the Kauner Valley. Except for the lapse rate between the first two eta levels, all the others up to level 20 range between -0.52 and -0.67°C/100 m. We will add this information to the manuscript.

We will perform additional evaluations regarding the selection of the eta level combinations and the spatial sensitivity for the computed lapse rates.

3. The main focus of the study is exploring the meteorological and model differences when using pressure-level lapse rate vs. near-surface WRF air temperatures. The title and main aims of the work are somewhat misleading therefore with respect to "impact of small-scale surface processes". I think that a more clear and specific title, as well as rephrased terminology throughout the manuscript is warranted. The manuscript really deals with the impact of correcting air temperatures due to coarse land surface representation of WRF.

Yes, we agree and will choose a more suitable title that avoids confusion and revise the terminology. In addition, we will clearly link the title, abstract, introduction, and synthesis to enhance the comprehensibility of this study.

4. The relevance of the corrected vs uncorrected air temperatures needs to be put into perspective when also adjusting and downscaling other meteorological variables (e.g. humidity) in a similar way. Describing their impact on long term snow and glacier evolution and objective ways to choose pressure levels/lapse rates are needed to be seen as a technical advance for the community. This would not require any rerunning over heavy WRF simulations (which would be excessive), but rather just the post-processing of corrections for inputs to WaSim.

We discussed the limitations of a single-variate correction, focusing solely on temperature in the discussion section. A multivariate bias correction would be more consistent from a physical perspective. We will double-check the WRF outputs to see whether relative humidity is also available for the individual eta levels. If so, we will perform additional evaluations and a multi-objective correction considering temperature and relative humidity. Otherwise, we can correct the WRF 2 m humidity based on the corrected WRF temperature.

**Technical Comments**

**Introduction**

The authors should reorder and restructure parts of the introduction for a more logical flow related to the aims of the study (the introduction starts talking about geomorphology and fluvial transport, and then moves onto droughts and flooding, which is too specific given the focus of the study). Given the change of article type, the authors should now go directly into the importance of accurate meteorology in hydrological models.

Yes, we agree and will restructure the introduction, focusing on the actual topic of accurate meteorology in hydrological models.

L42: This should be reworded. Observations since the LIA have only improved. Observations back in time to the LIA, however, are limited in space and time.

Yes, we agree and revise this sentence accordingly.

L44: The reference should be "van Tiel". Likely a citation manager issue.

We will correct the citation mistake.

L55: Given the specific application of WRF and a study about its limitations, the introduction and explicit benefits of HiCAR don't make sense here, especially as HiCAR is no longer used nor mentioned.

Yes, we agree and will remove the section about the HiCAR model.

L93-94: Correcting the diel cycle of what exactly? The reasoning of this paragraph needs to be made clearer.

In the past, bias corrections have mainly been applied to daily RCM data. However, the study of Faghih et al. (2022) found that even the diel cycle of variables simulated by RCMs is biased. This diel bias can impact the hydrological model results if not corrected.

L96: Small scale processes need to be clearly defined here for the reader. It seems that this only refers to the land cover representation from WRF, which is anyway largely removed and averaged in the manuscript's workflow to model at 25m. In essence, the small scale processes are not really considered, and the model is improved as a result. In line with my major comment above, this needs to be made clear from the title and re-wording of the manuscript.

We understand that the current wording contradicts the actual methodology of averaging these small-scale processes. We will rephrase the wording accordingly in this and other sections (title and abstract).

L96: RQ2 is not grammatically correct. It is written as a question, but the syntax is not correct.

We will rephrase RQ2.

Study Site

I guess that the data from Weissseespitze lies too far beyond the modelling period for this work? https://doi.pangaea.de/10.1594/PANGAEA.939830 This should be at least mentioned, explaining that site specific meteo and mass balance data are not available for the catchment of interest.

We are aware of the meteorological data of the Weissseespitze, and it would be very interesting to include them in our study. Unfortunately, there is no temporal overlap with our WRF data, as the Weissseespitz station has been in operation since 2017. Nevertheless, we will mention the data for completeness.

L130: Why is the Vergoetschen station not listed?

Table 1 only lists the temperature records used in this study for evaluating the temperature correction. But we can also add the Vergoetschen station to this table.

L136-140: Please describe (in brief) the SWE model. What are the key, perhaps site-specific, parameters?

We will add more information to section 2.3.1 regarding the calculation of SWE data.

L186-187: Provide an indication of the seasonality for the strength of the surface cooling effect over ice and snow. Is this related to the albedo and radiative cooling during winter primarily? Or also the summer, density-driven cooling over ice due to glacier wind development? Are there notable differences here due to ice vs. snow? What is responsible for the anomalous 2m temperatures over shrubland (Fig. 2a)? Does it meaningfully influence the lapse rate derivation if not averaging out the spatial effects (equation 1)?

It is an interesting point to investigate the seasonal cooling effects over snow and ice. We have to double-check the WRF outputs to determine whether we can perform a detailed analysis on that topic. As the WRF land cover classes do not differentiate between glaciers or permanent snow, we cannot investigate differences in the cooling effect due to albedo and density.

Fig. 2a actually shows the surface (or skin) temperature (TSK) and not the 2 m temperature. The anomaly of one shrubland grid cell is challenging to explain. It may be the exposition of this particular grid cell. We will double-check the topographical characteristics of the cells. Nevertheless, this anomaly does not affect the computation of the temperature lapse rates, as we are not considering the TSK.

L188: Define what is meant by "lower gradients" Which gradient exactly? Does lower mean a shallower lapse rate (a smaller change of temperature with elevation)?

Yes, lower gradients mean a smaller change of temperature with elevation in this context. We will clarify this.

L204: Here the authors are defining T, not TlevelX, I believe. Please check, in case that this is a typo.

Indeed, this is a typo. It should be T(t,i,j). We will correct that.

L209: As per my major comment, is this actually more robust? And more robust in what way? Evaluated how? The authors provide no evidence of this. This needs to be more rigorously tested and demonstrated for the reader.

As mentioned above, the eta levels are less dependent on the WRF land cover types when computing the spatial average, which depends on the catchment domain. We will rephrase this sentence and perform additional evaluations on this topic.

L214: The exact justification for the use of a linear gradient between levels 1 and 20 is not so solid from my reading. Levels 20-25 could also represent the mountain boundary layer. Can the authors provide some generalisable means of excluding the free-air based upon an objective measure? For example, how strong is the relationship of air temperature and elevation (e.g. the R2 of the levels in Fig. 2). More importantly, does it make any difference? How would another study utilise this information (which forms the core part of the manuscript) to improve their own hydrological model simulations? What impact does ignoring the lowest eta have?

We agree that there is no scientific justification for the choice of eta level range from 1 to 20. We will perform additional calculations to investigate the sensitivity of the temperature lapse rate up to level 25. As suggested, we will compute the R2 between the four observation stations and different eta level

combinations. However, we are not expecting to see significant differences between the lapse rates computed for the eta level range 1 to 20 or 1 to 25, as eta level 20 corresponds to a mean elevation of about 6000 m a.s.l., which is far higher than the highest peaks (~ 3700 m a.s.l.) of the research area. On the contrary, ignoring the lowest eta level in the computation of the temperature lapse rate will likely have a larger impact that can affect the hydrological model results.

Fig3: The exact derivation of the pressure level lapse rate for this figure is unclear. Is this the regression of temperature and elevation for each level, as shown by Fig 2? This approach is however different to how the authors derive lapse rates applied to WaSim, however (by averaging temperatures for each level). What about taking the lapse rate in a similar way to what we see in Fig 3, from the nearest WRF pixel? This would implicitly account for the often shallower lapse rates observed over glaciers. Regarding this figure, the distinction to the methodological approach of the text should be given and explained. Ideally, the authors can also add the approximate elevation from the eta levels for interpretability, marking the elevation range of the catchment on top.

As mentioned above, the calculation of the lapse rates shown in Figure 1 (Fig. 3 in the manuscript) is based on two pairs of consecutive eta levels (e.g., 1 and 2). Accordingly, it is the mean lapse rate of two consecutive eta levels and of all grid cells in the Kauner Valley. We will add this information to the manuscript. This figure is intended to illustrate the sensitivity of the temperature lapse rate to the eta levels. We will highlight the elevation range of the research area in Figure 3 of the manuscript.

In principle, it would be conceivable to calculate the temperature lapse rate from just one WRF grid cell. However, there is a risk that the TLR would then be overestimated or underestimated, depending on the WRF land cover class. This approach may be effective for a small research area of a few square kilometers. However, for a larger study area (> 10 km²) with significantly different WRF land cover classes, it would lead to a distortion of the TLR. We therefore calculate the mean over the entire research area. We will add additional analysis regarding land cover dependent TLRs to the Supplement to strengthen our decision.

L237: So the complex data results in a single lapse rate value to correct the (2m?) WRF cell to the elevation of the meteo station? This seems a huge simplification of a rich data input from WRF, but likely also depends on the absolute differences of elevation between the two. These elevation differences (HWRF and HStation) should be given somewhere.

We will add a table showing the elevation differences between the WRF grid cell and the meteorological station to the Supplement. In theory, we could also use the hourly lapse rates from Eq. 3 for a more accurate representation of the TLR. Nevertheless, it is unlikely to impact the hydrological model results, as the temperature values computed with Eq. 6 are only considered in the comparison of the ECDFs in Figure 4 of the manuscript.

Section 2.4.2: The precipitation frequencies can be highly influential to high resolution, physically-oriented models, so I am happy to see that this has been considered. For the re-structuring of the manuscript, I think these are details which can be considered away from the main text somehow, however. Are there other bias-corrections necessary to improve the representation of glacier mass

balance in this catchment? Related to this, What comparison to geodetic glacier changes has been made, and how sensitive would the model results (i.e. Fig. 7d) be to variations of the lapse rate derivation as I have suggested should be considered?

We appreciate your positive feedback regarding the hourly frequency correction of the WRF precipitation. We also believe that this correction is valuable for physics-based snow-hydrological simulations with an hourly time step. We would put this section in the supplement when necessary to further reduce the manuscript's length.

**Results**

Fig. 4: Having some statistics for the improvement of fit would be beneficial here (mean bias, RMSE etc).

We listed some information on the ECDFs, which are computed using the KS test, in the Supplement (Table S1) for comparing the statistical characteristics. We can add another table listing some goodness-of-fit criteria, such as mean bias or RMSE.

L455-456: Citation for this statement is necessary.

We will add a reference for this statement.

Fig. 8 a,c,e,g: The standard deviation of what? The sub-daily simulated values? This should be clarified.

Yes, it's the sub-daily standard deviation for the day of the year over a particular period (e.g., 1973-1987). We will clarify this in the caption.

Fig 9: These are nice results and quite valuable for the community, but they seem to address a broader question about the long-term evolution of the ice and snow in this catchment, but not really related to the value of the authors approach to improve the temperature downscaling. This could be left for successive work on this topic, or revised to demonstrate the long-term implication of different approaches for changing/improving/simplifying WRF's surface representation. Also for this figure, the authors should specify whether these are hydrological years or not. I suspect they are, as 2002 has the lowest snow cover and not 2003 (the European heatwave).

We agree that Figure 9 is valuable for demonstrating the plausibility of the hydrological simulation results with the corrected WRF temperature over a longer period. Yes, the years are illustrated as hydrological years. We will add this information.

**Discussion**

The authors have pre-empted the critique of their modelling approach that has overlooked humidity and other variables which are also dependent on WRF's coarse (and perhaps incorrect) surface characteristics. Nevertheless for a physical modelling approach, the accurate estimation of most or all meteorological variables can be quite important. The relevance of the improvements to the air temperature downscaling cannot be put into perspective, however. I think that upon revising and restructuring the manuscript as a technical note, some testing of the different downscaling of the

humidity and precipitation should be made to highlight whether it is as influential to the model calculations as the corrections made to air temperature.

We will conduct more detailed evaluations of the precipitation and humidity corrections to test whether these corrections have an influence on the model results.

L606: Why did the authors decide upon the usage of 20CRv3 when ERA5 is a widely adopted reanalysis, especially in Europe. This should be mentioned here.

We have decided to use the 20CRv3 reanalysis data, as this data is also available for the 19th century. Therefore, this dataset enables the investigation of long-term cryo-hydrological changes in the European Alps since the Last Little Ice Age in 1850. We agree that the ERA5 reanalysis is widely used in Europe. However, this data set is available from 1940. We will elaborate more about the reasoning.

L614: Can the authors justify (ideally early in the manuscript) why a very high resolution of 25m was chosen or necessary.

The 25 m grid resolution of the hydrological model is a compromise between the spatial discretization needed to resolve the small-scale heterogeneity of the research area ( $\sim 60 \text{ km}^2 \text{ in size}$ ) with a complex topography and the computational time needed to run the hydrological model in hourly time steps for longer periods ( $\sim 40 \text{ years}$ ). We will clarify the selection of the model discretization in the manuscript.

**References**

Faghih, M., Brissette, F., and Sabeti, P.: Impact of correcting sub-daily climate model biases for hydrological studies, Hydrology and Earth System Sciences, 26, 1545–1563, https://doi.org/10.5194/hess-26-1545-2022, 2022